# Inhibition of mitochondrial protein import and proteostasis by a pro-apoptotic lipid

Josep Fita-Torró[1], José Luis Garrido-Huarte[1], Lucía López-Gil[1], Agnès H Michel[2], Benoit Kornmann[2], Amparo Pascual-Ahuir[3]*, Markus Proft[1]*

[1]Department of Metabolism, Inflammation and Aging, Instituto de Biomedicina de Valencia IBV-CSIC; Valencia Biomedical Research Foundation Centro de Investigación Príncipe Felipe (CIPF) – Associated Unit to the Instituto de Biomedicina de Valencia IBV-CSIC, Valencia, Spain; [2]Department of Biochemistry, University of Oxford, Oxford, United Kingdom; [3]Grupo de Ingeniería Biomolecular y Biosensores, Centro de Investigación e Innovación en Bioingeniería Ci2B, Universitat Politècnica de València, Ciudad Politécnica de la Innovación, Valencia, Spain

## eLife Assessment

This study is a **valuable** observation that deals with the toxic effects of an intermediary in lipid degradation [trans-2-hexadecenal (t-2-hex)] in yeast through modification of mitochondrial protein import via the TOM complex. We find that the claim that the TOM complex is a main target of t-2-hex are supported by **solid** evidence, however the molecular mechanism remains unclear allowing multiple interpretation. Despite the shortcomings, this study is inspiring for researchers from the organellar, protein trafficking and lipid field and serves as a starting point to further precise and mechanistic analyses of the phenomenon.

*For correspondence:
apascual@ci2b.upv.es (AP-A);
mproft@ibv.csic.es (MP)

**Abstract** Mitochondria-mediated cell death is critically regulated by bioactive lipids derived from sphingolipid metabolism. The lipid aldehyde trans-2-hexadecenal (t-2-hex) induces mitochondrial dysfunction from yeast to humans. Here, we apply unbiased transcriptomic, functional genomics, and chemoproteomic approaches in the yeast model to uncover the principal mechanisms and biological targets underlying this lipid-induced mitochondrial inhibition. We find that loss of Hfd1 fatty aldehyde dehydrogenase function efficiently sensitizes cells for t-2-hex inhibition and apoptotic cell death. Excess of t-2-hex causes a profound transcriptomic response with characteristic hallmarks of impaired mitochondrial protein import, like activation of mitochondrial and cytosolic chaperones or proteasomal function and severe repression of translation. We confirm that t-2-hex stress induces rapid accumulation of mitochondrial pre-proteins and protein aggregates and subsequent activation of Hsf1- and Rpn4-dependent gene expression. By saturated transposon mutagenesis, we find that t-2-hex tolerance requires an efficient heat shock response and specific mitochondrial and ER functions and that mutations in ribosome, protein, and amino acid biogenesis are beneficial upon t-2-hex stress. We further show that genetic and pharmacological inhibition of protein translation causes t-2-hex resistance, indicating that loss of proteostasis is the predominant consequence of the pro-apoptotic lipid. Several TOM subunits, including the central Tom40 channel, are lipidated by t-2-hex in vitro and mutation of accessory subunits Tom20 or Tom70 confers t-2-hex tolerance. Moreover, the Hfd1 gene dose determines the strength of t-2-hex mediated inhibition of mitochondrial protein import, and Hfd1 co-purifies with Tom70. Our results indicate that the transport of mitochondrial precursor proteins through the outer mitochondrial membrane is sensitively inhibited by the pro-apoptotic lipid and thus represents a hotspot for pro- and anti-apoptotic signaling.

## Introduction

Mitochondria are universal and essential organelles, which beyond fundamental functions in energy metabolism, the biogenesis of Fe/S clusters, and $Ca^{2+}$ homeostasis, are the central regulators of the most common form of programmed cell death found in eukaryotic cells (*Bock and Tait, 2020*; *Chipuk et al., 2021*). Consequently, mitochondrial function and dysfunction critically establish cell survival or deterioration during aging, neurodegeneration, and cancer (*Area-Gomez et al., 2019*; *Porporato et al., 2018*; *Son and Lee, 2021*). In this process of intrinsic apoptosis, cellular stress triggers pro-death signals, which converge at the organelle and induce the permeabilization of the outer mitochondrial membrane (MOMP). As a consequence, the release of several soluble mitochondrial proteins such as cytochrome *c* from the intermembrane space into the cytosol executes programmed cell death via caspase-dependent and –independent routes. Because MOMP is the earliest irreversible death trigger, its misregulation is causally linked to many diseases and is a promising target for therapeutic interventions (*Jeng et al., 2018*). Accordingly, the mitochondria-mediated death decision needs to be strictly controlled. In higher eukaryotes, it is known that the entry into the death program depends on the functional interplay of different members of the BCL-2 protein family (*Jeng et al., 2018*). Specifically, the BCL-2 protein BAX can transform under stress conditions from a latent cytosolic form to the pro-apoptotic oligomeric form, which initiates the perforation of the outer mitochondrial membrane and commits the cell to death (*Walensky and Gavathiotis, 2011*). One important induction mechanism of mitochondrial cell death is triggered by lipids derived from sphingolipid metabolism at the endoplasmic reticulum. Mitochondria devoid of ER membranes are resistant to pro-apoptotic inhibition, and specific lipid species alone, such as sphingosine-1-phosphate or trans-2-hexadecenal (t-2-hex), efficiently activate Bax-driven MOMP (*Chipuk et al., 2012*). The pro-apoptotic function of externally added t-2-hex has been generally confirmed in cultures from yeast to mammalian cells (*Amaegberi et al., 2019*; *Kumar et al., 2011*; *Manzanares-Estreder et al., 2017*). One explicit pro-apoptotic mechanism of t-2-hex has been recently discovered, consisting of the direct lipidation and subsequent activation of Bax at C126 (*Cohen et al., 2020*). However, t-2-hex covalently and specifically modifies hundreds of different human proteins in vitro (*Jarugumilli et al., 2018*), which motivated our present study to identify biologically important t-2-hex targets in an exhaustive manner.

99% of the mitochondrial proteins are synthesized at cytosolic ribosomes and thus have to be imported into the organelle mostly post-translationally (*Pfanner et al., 2019*). This large variety of precursor proteins needs to pass through the mitochondrial membrane(s), be sent to the correct mitochondrial subcompartment, be correctly folded, and eventually assembled into protein complexes. Mitochondrial protein import is a proteostatic challenge for the cell and recent research has revealed that multiple quality control mechanisms exist to avoid the accumulation of non-imported or aggregated precursor proteins (*Song et al., 2021*). The multisubunit TOM complex (translocase of the outer membrane) is the entry channel for the vast majority of mitochondrial precursors. The β-barrel Tom40 protein forms the actual protein-transporting pore at the outer mitochondrial membrane (*Hill et al., 1998*). The majority of the mitochondrial pre-proteins made in the cytosol have to be guided and maintained in an unfolded, import-competent state, which critically requires the function of cytosolic chaperones, such as Hsp70 and Hsp90 (*Young et al., 2003*). This is important for cell homeostasis as prematurely folded pre-proteins cannot pass through and eventually clog the TOM complex, which leads to mitochondrial dysfunction and triggers a strong stress response (*Boos et al., 2019*). Cellular rescue pathways, which prevent clogging of the TOM complex, have been recently discovered in yeast. The mitochondrial protein translocation-associated degradation pathway (mito TAD) continuously monitors protein import at the TOM complex upon normal growth conditions and removes stalled pre-proteins with the help of the ubiquitin-binding protein Ubx2 and the AAA protein unfoldase Cdc48 by facilitating their proteasomal degradation (*Mårtensson et al., 2019*). Whenever mitochondrial protein import is diminished by either mutation of translocase subunits, overexpression of clogging-prone pre-proteins or a loss of mitochondrial membrane potential, the mitochondrial compromised protein import response (mitoCPR) is activated. Upon prolonged mitochondrial import stress, the stress-induced Cis1 protein recruits the Msp1 AAA ATPase at the TOM complex to remove arrested precursor proteins by proteasomal degradation (*Weidberg and Amon, 2018*). In situations when these internal rescue systems are overwhelmed, the accumulation of non-imported mitochondrial precursor proteins in the cytosol poses a serious threat to cellular proteostasis and causes a strong proteotoxic stress (*Wang and Chen, 2015*; *Wrobel et al., 2015*). Under these conditions, the

cell responds with a massive transcriptional remodeling, which involves heat shock factor 1 (Hsf1) and Rpn4 in order to induce the expression of cytosolic chaperones and the ubiquitin-proteasome system, respectively (*Boos et al., 2019*). The efficiency of mitochondrial protein import is of central importance for general cytosolic protein folding and homeostasis (*Boos et al., 2020*). It has been very recently demonstrated that the toxic aggregation of polyQ, amyloid β, and α-synuclein proteins relevant for neurodegenerative and aging phenotypes is dependent on the efficiency of mitochondrial protein import (*Nowicka et al., 2021Schlagowski et al., 2021*). Here, we demonstrate that mitochondrial protein import is inhibited by the naturally occurring pro-apoptotic sphingolipid degradation intermediate t-2-hex, the accumulation of which leads to a massive proteostatic response in the cytosol to prevent general protein aggregation. Our results connect pro-apoptotic lipid function with protein misfolding and aggregation via a block in mitochondrial protein import.

## Results

### The stress-regulated fatty aldehyde dehydrogenase Hfd1 is the major detoxifier of pro-apoptotic t-2-hex

Intracellular production of the pro-apoptotic lipid t-2-hex occurs in eukaryotic cells via the evolutionarily conserved sphingolipid degradation pathway (*Figure 1A*). In human cells, t-2-hex causes mitochondrial dysfunction by directly stimulating Bax oligomerization at the outer mitochondrial membrane. In yeast, however, t-2-hex efficiently interferes with mitochondrial function and cell growth in a Bax-independent manner (*Manzanares-Estreder et al., 2017*). Thus, we wanted to identify novel lipid-mediated apoptotic mechanisms in the yeast model. In *S. cerevisiae*, the degradation of sphingosine to palmitic acid occurs through enzymatic steps, which seem to be transcriptionally activated upon cellular stress (*Manzanares-Estreder et al., 2017*). The fatty aldehyde dehydrogenase Hfd1 acts as a potential direct detoxifier by oxidizing t-2-hex. We analyzed the regulation of *HFD1* by cytotoxic stresses with time-elapsed luciferase assays in vivo. We applied different treatments including NaCl for the induction of osmotic stress, menadione for the induction of oxidative stress (*Flattery-O'Brien et al., 1993*), and acetic acid for the induction of pro-apoptotic stress (*Chaves et al., 2021*). Indeed, *HFD1* expression was fast, robustly, and transiently induced by the three types of stress (*Figure 1B*). The sensitivity of the observed activation was comparable to that of a generic stress-inducible gene, *GRE2*, except for oxidative stress, where *HFD1* showed a less sensitive response (*Figure 1C*). This suggested that the yeast sphingolipid degradation pathway is generally induced by diverse stresses and that increased Hfd1 levels might be necessary to reduce toxic t-2-hex levels under all these conditions. We next asked whether increased t-2-hex levels itself triggered at all a transcriptional stress response in yeast cells. External addition of the bioactive lipid indeed caused a dose-dependent activation of the *HFD1* and *GRE2* genes (*Figure 1D*). Of note, in contrast to the transient stimulation of gene expression caused by 'conventional' abiotic stresses (*Figure 1B*), t-2-hex overload activated gene expression in a much more sustained manner. This suggested that the lipid caused intracellular damage, which builds up slowly and affects cell function over a prolonged time. We tested whether the amount of Hfd1 enzyme was critical for the extent of t-2-hex inflicted damage. We compared how the sensitivity to the lipid changed upon absent or constitutively high Hfd1 activity. *hfd1Δ* cells exhibited a very pronounced growth delay upon t-2-hex treatment, while constitutively Hfd1 overexpressing cells gained robust growth upon the same conditions (*Figure 1E*). These results confirmed that *HFD1* gene dose and enzyme activity were critical determinants for the tolerance to the pro-apoptotic lipid. Finally, we found that the loss of Hfd1 function greatly enhanced the adaptive activation of gene expression (*Figure 1F*). Furthermore, *hfd1* mutants proliferated less efficiently upon pro-apoptotic stress conditions, while cells with constitutive *HFD1* were slightly resistant (*Figure 1G*). These data suggest that the growth arrest by pro-apoptotic t-2-hex is modulated by the efficiency of internal t-2-hex detoxification, which depends on the Hfd1 enzyme levels. *hfd1Δ* cells are highly sensitized to t-2-hex toxicity, exhibiting a several-fold enhanced response, a fact that will be exploited next to discover the complete transcriptomic response to the pro-apoptotic lipid.

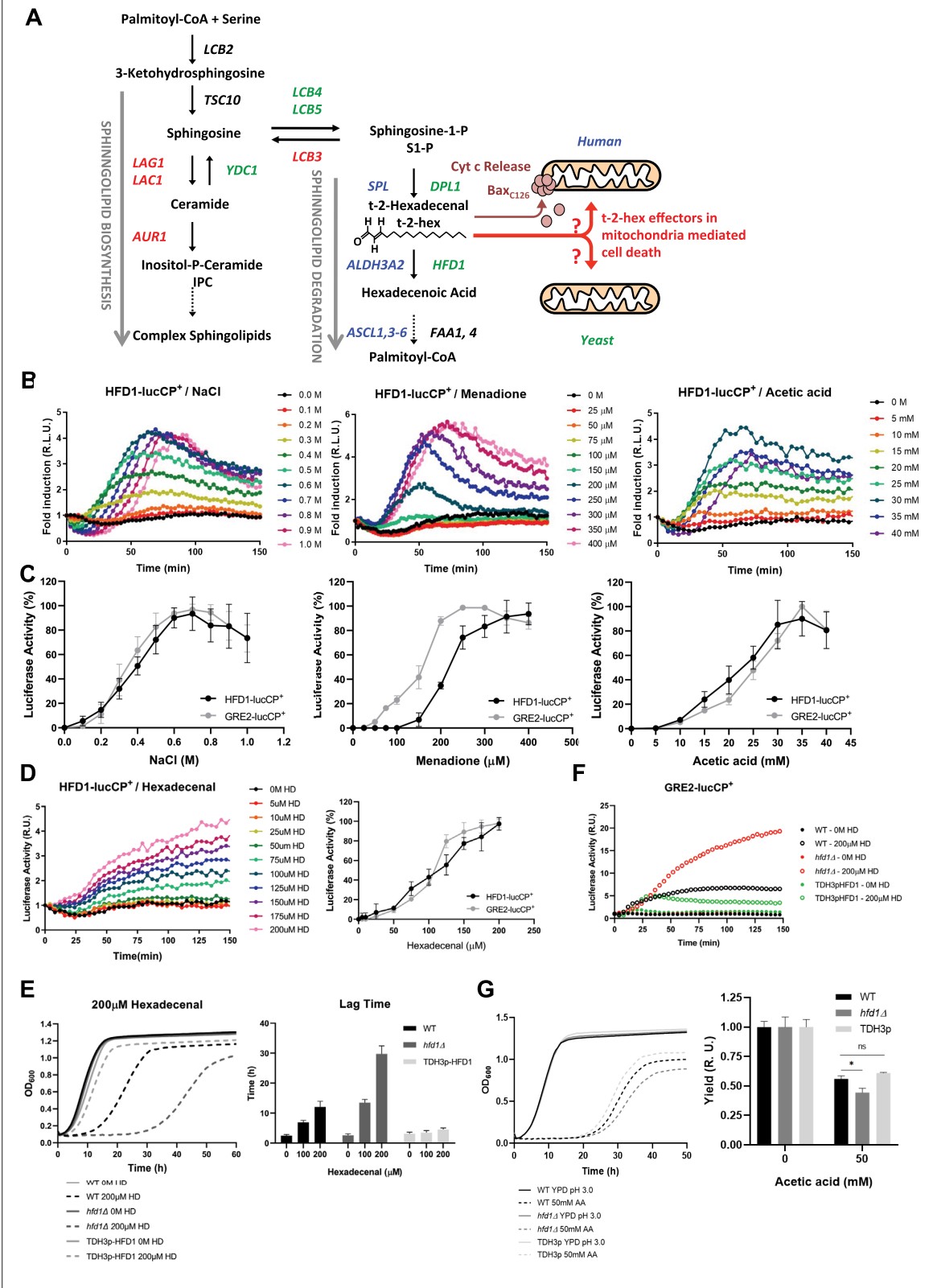

**Figure 1.** The stress-activated fatty aldehyde dehydrogenase Hfd1 determines trans-2-hexadecenal (t-2-hex) mediated toxicity and adaptive response. (**A**) Overview of the generation of t-2-hex within the sphingolipid degradation pathway. Stress-activated (-repressed) enzymatic functions are depicted in green (red) for yeast. The corresponding human enzymes involved in t-2-hex chemistry are shown in blue. t-2-hex stimulates the pro-apoptotic activity of Bax at human mitochondria by $C_{126}$ lipidation. (**B**) *HFD1* expression is activated by different cytotoxic stresses. Hyperosmotic (NaCl), oxidative

*Figure 1 continued on next page*

*Figure 1 continued*

(Menadione), and pro-apoptotic (Acetic acid) stresses were gradually applied to *HFD1p*-luciferase reporter containing yeast cells. The relative luciferase activity was measured in vivo (n=3). (**C**) Comparison of different stress sensitivities of the *HFD1* and *GRE2* promoters. (**D**) External addition of t-2-hex (HD) activates *HFD1* expression in a concentration-dependent manner. Upper panel: *HFD1*-luciferase activation upon gradual addition of t-2-hex (n=3). Lower panel: Comparison of *HFD1* and *GRE2* activation sensitivities to t-2-hex. (**E**) Left panel: *HFD1* gene dose modulates susceptibility to t-2-hex. Growth curves of wild-type, *hfd1Δ*, and constitutively Hfd1 overexpressing strains (*TDH3p-HFD1*) in the absence and presence of t-2-hex (n=3). Right panel: Quantitative comparison of growth performance (length of lag phase) of the same strains upon the indicated t-2-hex concentrations (n=3). (**F**) Loss of Hfd1 function causes a hypersensitive t-2-hex response. *GRE2p*-luciferase reporter assay in strains with the indicated *HFD1* gene dose in response to t-2-hex (n=3). (**G**) *HFD1* gene dose modulates the tolerance to pro-apoptotic concentrations of acetic acid (n=3). *p<0.05 by Student´s unpaired t-test.

## The transcriptomic response to t-2-hex reveals global features of proteostatic imbalance

Our results indicated that cells lacking Hfd1 activity remain for many hours in a non-proliferative state when challenged with a t-2-hex overload. In this period of time, a strong regulation of gene expression occurs, presumably in order to prepare defense mechanisms against the toxic effects of the lipid. We were interested in capturing as many aspects as possible of this adaptive response and determined the transcriptomic changes upon heavy t-2-hex overload in *hfd1Δ* cells by RNA-seq (*Figure 2A*). We found that adaptation to the pro-apoptotic lipid involved the profound remodeling of the yeast transcriptome. The expression of approximately 30% of the transcriptome was more than threefold up- or down-regulated upon t-2-hex stress (*Supplementary files 1 and 2*). Very strong (>10 fold) activation was observed for a remarkably high number of protein-encoding genes (263), which were preferentially distributed in categories such as 'stress-related enzyme function,' 'mitochondrial function,' 'transport function,' or 'chaperones.' The most repressed gene functions fell into the categories 'ribosome and translation,' 'pheromone response and mating,' 'nucleotide biosynthesis,' or 'cell cycle.' Next, genes up-regulated (>2 log$_2$FC) or down-regulated (<-2 log$_2$FC) by t-2-hex were selected and subjected to GO category enrichment analysis (up-regulated, *Supplementary file 1*; down-regulated, *Supplementary file 2*). We found that 'Mitochondrial organization' was the most numerous GO group activated by t-2-hex, while it was 'Ribosomal subunit biogenesis' for t-2-hex repression. We, therefore, analyzed the most strongly t-2-hex induced genes encoding mitochondrial proteins (*Figure 2B*). Remarkably, *CIS1* showed an extraordinarily high induction (>250 fold) and thus represented the most inducible gene with a known mitochondrial function. Cis1 is the central protein of the MitoCPR pathway, which is specifically induced upon and protects from mitochondrial protein import stress, together with the Msp1 AAA-ATPase (*Weidberg and Amon, 2018*). *MSP1* expression, although more moderately, is also significantly induced upon t-2-hex stress. These data suggested that mitochondrial protein import might be impaired by the action of the pro-apoptotic lipid. Accordingly, we found the expression of a large group of mitochondrial chaperones to be significantly activated upon t-2-hex exposure (*Figure 2B*). We compared our t-2-hex transcriptomic response with the previously described specific response to mitochondrial protein import block (*Weidberg and Amon, 2018*), taking into account only robustly (> threefold) induced genes. We found that 69% of all genes induced by mitochondrial import stress were also positively affected by t-2-hex stress (*Supplementary file 3*). The number of coinciding genes increased to 85% when we exclusively compared mitochondrial functions in both datasets, indicating that the response to the bioactive lipid included a specific adaptation to a failure in the mitochondrial protein import.

The accumulation of non-imported mitochondrial proteins triggers a global transcriptional proteostatic response (*Boos et al., 2019*). We, therefore, investigated whether the t-2-hex response contained general features of this proteostatic adaptation. Strikingly, the expression of many cytosolic chaperone systems of Hsp70, Hsp90, Hsp40, and other groups was highly activated by t-2-hex overload (*Figure 2C*), with some chaperone genes showing extraordinary induction levels of 100–1000-fold. Notably, these chaperones are key players in the maintenance of proteostasis in the cell (*Sontag et al., 2017*). The expression of other chaperone systems, such as the Cct chaperonin or the prefoldin complexes, dedicated to the correct assembly of cytoskeleton components, is not significantly affected by the lipid aldehyde (*Figure 2C*). Additionally, we observed a remarkably homogeneous upregulation of the expression of all proteasomal subunits and strong activation of positive proteasomal regulators, such as the Rpn4 transcriptional activator of proteasomal genes or the Roq1,

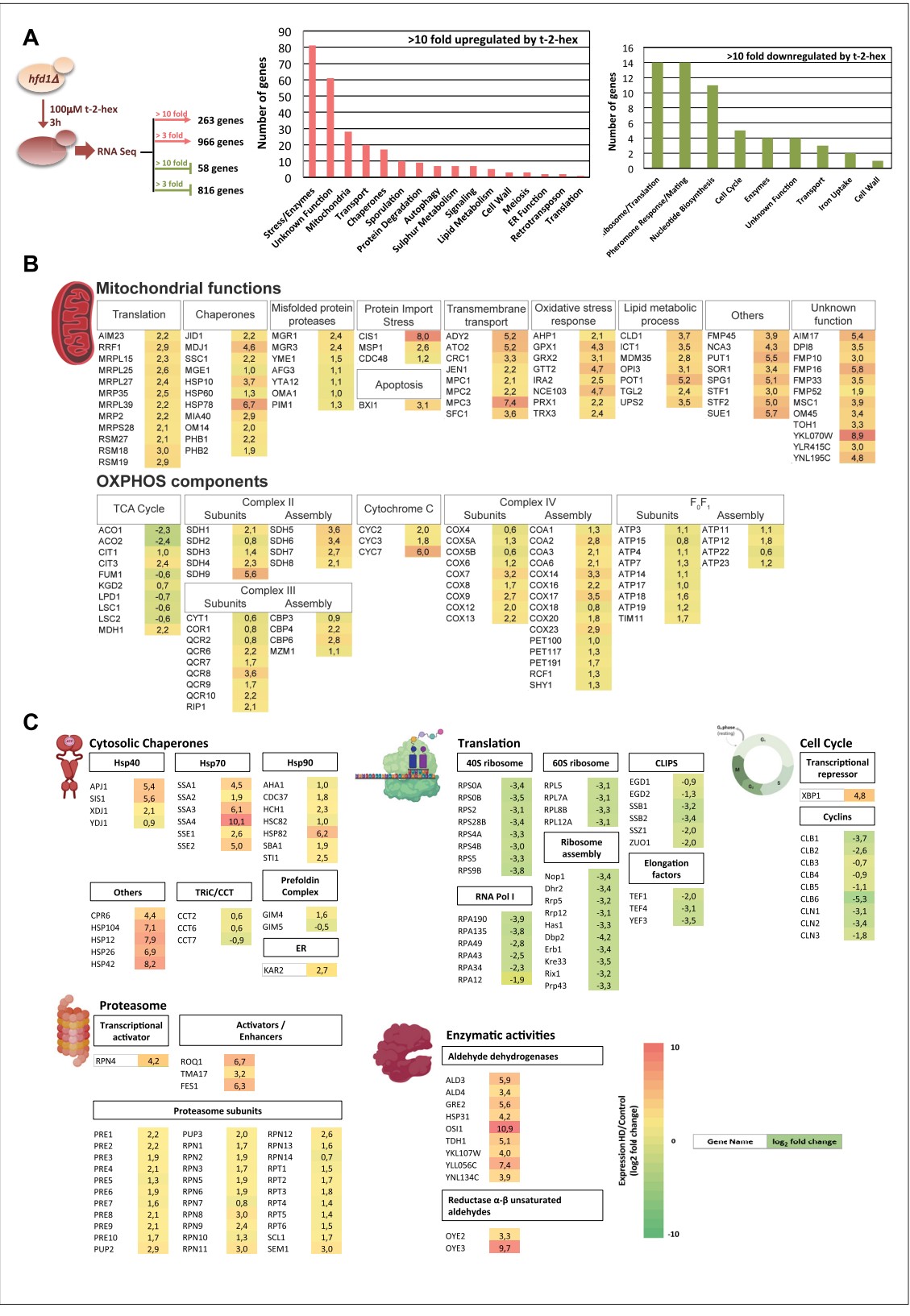

**Figure 2.** Trans-2-hexadecenal (t-2-hex) stress causes profound transcriptomic remodeling. (**A**) Transcriptomic analysis of the cellular response to t-2-hex overload. RNA seq experiment setup and general functional groups of up- and down-regulated genes. (**B**) t-2-hex activated mitochondrial functions. (**C**) Genomic remodeling of gene expression reveals a general proteostatic response of the cell to t-2-hex. Cytosolic chaperones are strongly activated in general, but not the Cct chaperonin or the prefoldin complex. Structural components and regulators of the proteasome are coordinately up-regulated,

*Figure 2 continued on next page*

*Figure 2 continued*

while the cytosolic translation machinery and cell cycle regulators are strongly repressed. Significantly up-regulated enzymes with known aldehyde dehydrogenase or unsaturated aldehyde reductase activities are shown. Colors indicate $\log_2$ fold changes in gene expression of t-2-hex treated versus mock-treated cells (means of n=3 independent biological replicates) for genes representing selected functional groups or complexes.

Fes1, and Tma17 stimulators of proteasomal assembly and activity (*Gowda et al., 2013*; *Hanssum et al., 2014*; *Szoradi et al., 2018*). In parallel, our RNA seq data revealed that t-2-hex repressed very strongly the expression of genes involved in the de novo biosynthesis of ribosomes and proteins. This inhibition was observed in a highly consistent manner for specific RNA polymerase I and ribosomal subunits, as well as ribosome assembly factors, translational elongation factors, and co-translationally acting chaperones (CLIPs). These data indicated that adaptation to the pro-apoptotic t-2-hex involved the activation of cytosolic chaperone systems and the proteasomal protein degradation machinery, as well as a general repression of translation, presumably as a consequence of imbalances in the cytosolic proteostasis caused by impaired mitochondrial protein import. This seems to be accompanied by a general arrest of cell cycle progression, because we observed strong repression of all cyclin genes and simultaneous up-regulation of the Xbp1 transcriptional master regulator of quiescence and cell cycle repression (*Figure 2C*).

In order to further characterize the gene groups, which were most significantly affected by t-2-hex stress, we examined the significantly up- and down-regulated promoters by in silico analysis for enriched transcription factor binding motifs. Within the t-2-hex induced promoter regions, we found a consistent and strong enrichment of binding sites for Hsf1 and Rpn4 (*Figure 3A*). In yeast, Hsf1 is the essential master regulator of proteostasis, controlling both basal and heat stress-induced expression of many chaperone-encoding genes by binding to heat shock promoter elements (HSE) (*Akerfelt et al., 2010*). Rpn4 is a non-essential, general activator of proteasome genes, which binds to so-called proteasome-associated control elements (PACE) (*Xie and Varshavsky, 2001*). In turn, for the t-2-hex-repressed gene promoters, we identified the binding motifs for Stb3, Sfp1, and Rap1 as the most significantly enriched regulatory elements (*Figure 3A*). Stb3 is a general activator of yeast ribosomal RNA genes and binds to the ribosomal RNA processing elements (RRPE) (*Liko et al., 2007*). Sfp1 and Rap1 stimulate the expression of ribosomal protein and biogenesis genes (*Lieb et al., 2001Marion et al., 2004*). These data suggested that t-2-hex triggered the upregulation of chaperone and proteasome genes via the Hsf1 and Rpn4 transcriptional activators as the main adaptive response, while it massively repressed ribosomal biogenesis. We wanted to experimentally prove the prevalence of the heat shock and proteasomal response in the case of t-2-hex pro-apoptotic stimulation. To this end, we constructed a set of live cell reporters expressing destabilized luciferase (*Figure 3B*) under the control of transcription factor-specific response elements. We included binding sites for the Hsf1 and Rpn4 factors responding to protein aggregation and misfolding (HSE and PACE, respectively), the Pdr1/3 factors responding to xenobiotic stress (PDRE), and the Yap (Yap1 and other Yap TFs) and Sko1 factors responding to oxidative stress (AP-1 and CRE, respectively). We quantified the dose-response of each reporter separately upon continuous increase of t-2-hex concentrations by time-elapsed luciferase measurements. While all transcriptional reporters were robustly activated by the bioactive lipid, we found important differences in the sensitivity of the response for the different stress types. The Hsf1- and Rpn4-mediated transcriptional regulation showed the most sensitive activation upon t-2-hex stress with the lowest $EC_{50}$ values of 8 and 10 µM, respectively (*Figure 3C and D*). In contrast, the oxidation-specific reporters were activated with significantly lower sensitivity. Importantly, very low t-2-hex concentrations (10 µM) added from the outside to yeast wild-type cells induced a significant up-regulation of PACE-dependent gene expression (*Figure 3E*). These results confirmed that the transcriptional heat shock and proteasomal responses were preferentially triggered by the pro-apoptotic lipid at low µM concentrations, presumably because it caused an important problem of protein misfolding in the cell.

## Pro-apoptotic t-2-hex causes cytosolic protein aggregation

t-2-hex is an α/β unsaturated lipid aldehyde. In mammalian cells, t-2-hex exerts specific apoptotic functions by nucleophilic addition and direct lipidation of sulfhydryl groups of cysteines in the Bax protein (*Cohen et al., 2020*). We wanted to discern the biological effects of t-2-hex in yeast by comparing it to the saturated t-2-hex analog hexadecanal or t-2-hex-$H_2$. It is specifically missing the ability to perform

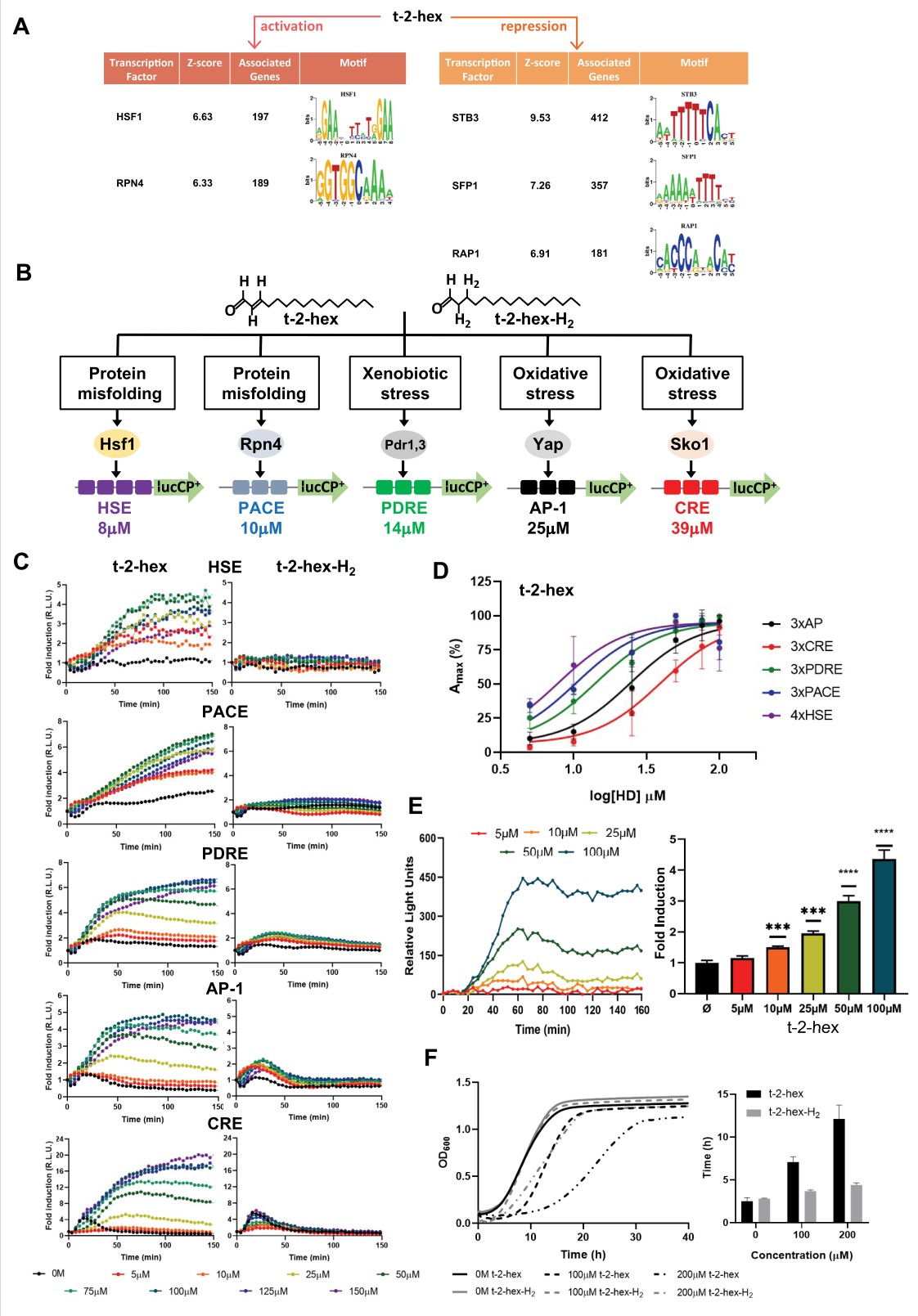

**Figure 3.** Trans-2-hexadecenal (t-2-hex) specifically induces the heat shock and proteasomal transcriptional response. (**A**) Significantly enriched promoter motifs were identified from the transcriptionally up- and down-regulated genes identified by our RNA seq study upon t-2-hex stress. (**B**) Schematic representation of the stress-specific luciferase reporters applied in the dose-response experiments upon unsaturated t-2-hex or saturated analog t-2-hex-H₂. The t-2-hex dose causing half-maximal induction for each reporter is given in µM below the constructs. (**C**) Dose-response curves of

*Figure 3 continued on next page*

*Figure 3 continued*

the indicated live cell luciferase reporters upon increasing t-2-hex and t-2-hex-$H_2$ concentrations (n=3). All reporters were assayed in *hfd1Δ* cells. Initial light emission levels at time 0 were set to 1. (**D**) Comparison of the sensitivities of different stress type-specific responses to the pro-apoptotic t-2-hex. Experimental data from (**C**) were analyzed by plotting the maximal reporter activation against the log[t-2-hex] concentration. (**E**) Induction of a 3xPACE-lucCP$^+$ reporter in wild-type yeast cells upon the indicated t-2-hex doses. Left panel: Dose-response curves corrected for the mock-treated samples. Right panel: Maximal induction fold for each lipid dose tested (n=3). \*\*\*p<0.001, \*\*\*\*p<0.0005 by Student's unpaired t-test. (**F**) t-2-hex unsaturation is the cause of its severe growth inhibition. Growth of yeast wild-type cells was scored upon the indicated t-2-hex and t-2-hex-$H_2$ concentrations (upper panel) and the lag time was calculated (lower panel), n=3.

a nucleophilic addition and direct lipidation, therefore, an absence of response upon exposure to the saturated compound indicates it is the lipidation that is responsible for the biological effects of t-2-hex. We found that the lack of α/β unsaturation in the lipid aldehyde changed the spectrum and intensity of stress-responsive transcriptional activation according to our live cell luciferase reporter assays. t-2-hex-$H_2$ mainly triggered an antioxidant and xenobiotic stress response, while it did not activate the protein misfolding response through Hsf1 or Rpn4 (*Figure 3B and C*). Interestingly, t-2-hex-$H_2$ caused a typical transient activation of stress gene expression as compared to the sustained response triggered by t-2-hex. Accordingly, we found that t-2-hex-$H_2$ caused significantly shorter growth inhibition as compared to the unsaturated t-2-hex (*Figure 3F*). These data indicated that the pro-apoptotic t-2-hex induced a long-lasting proteostatic emergency in the cell, possibly by direct lipidation of effector proteins. We next wanted to know whether t-2-hex was able to induce a general protein folding and aggregation problem in the cytosol. We monitored the intracellular distribution of the protein chaperone Hsp104, which was tagged with green fluorescent protein (GFP). Hsp104, together with Hsp70, is a molecular disaggregase (*Glover and Lindquist, 1998*) and can be used to visualize the cytosolic accumulation of protein aggregates (*Erjavec et al., 2007*; *Jacobson et al., 2017*). The exposure to t-2-hex caused the appearance of multiple foci of protein aggregates in wild-type cells, which increased in number over the first 6 hr of lipid stress (*Figure 4A*). Larger protein aggregates were observed in cells with impaired proteasome function (*rpn4Δ*), indicating that proteasomal activity was needed to clear lipid-induced protein aggregates in the cytosol. Interestingly, we observed that t-2-hex-$H_2$ only caused a modest and transient protein aggregation within the first hour of exposition (*Figure 4A*). Thus, the massive protein aggregation, which mounts slowly over time, was specifically induced by the unsaturated t-2-hex molecule. Protein aggregation can occur at mature and/or newly synthesized proteins depending on the nature of the proteostatic stress. Accordingly, the application of the translation inhibitor cycloheximide revealed that Hsp104 foci are formed independently of protein synthesis in the case of heat stress, but their formation depends on protein synthesis in the case of arsenite stress (*Jacobson et al., 2012*). We found that cycloheximide pre-treatment of the cells completely abolished t-2-hex triggered protein aggregates (*Figure 4B*), indicating that the pro-apoptotic lipid caused misfolding of de novo synthesized proteins.

The ubiquitin-proteasome system is a main controller of cellular proteostasis (*Dikic, 2017*). Our transcriptomic data identified a uniform up-regulation of proteasomal subunits and activators upon t-2-hex stress. On the other hand, the proteasome cannot directly degrade aggregated proteins and instead is sensitively inhibited by aggregates (*Bence et al., 2001*; *Tyedmers et al., 2010*). To further explore the function of the proteasome during pro-apoptotic lipid stress, we determined the regulation of proteasomal activity. We measured the chymotrypsin-like activity of the proteasome in cell extracts using a synthetic substrate and found that t-2-hex, but not t-2-hex-$H_2$, severely diminished the activity of the proteasome in wild-type yeast cells to levels comparable with the proteasome-deficient *rpn4Δ* mutant (*Figure 4C*). We next quantified the protein levels of the proteasomal subunit Rpn8 and confirmed a continuous up-regulation upon exposure to t-2-hex (*Figure 4D*), thus confirming the RNA-seq results. These data might be explained by an inhibition of the proteasomal activity by extensive protein aggregates formed upon t-2-hex stress, which the cell might want to counteract by stimulated proteasomal gene expression. We further confirmed that *rpn4Δ* cells were hypersensitive to pro-apoptotic t-2-hex treatment (*Figure 4E*).

## Pro-apoptotic t-2-hex efficiently impairs mitochondrial protein import

Having established that pro-apoptotic lipid stress elicits a strong activation of mitochondrial protein import surveillance and general proteostatic rescue systems, we tested directly, whether t-2-hex

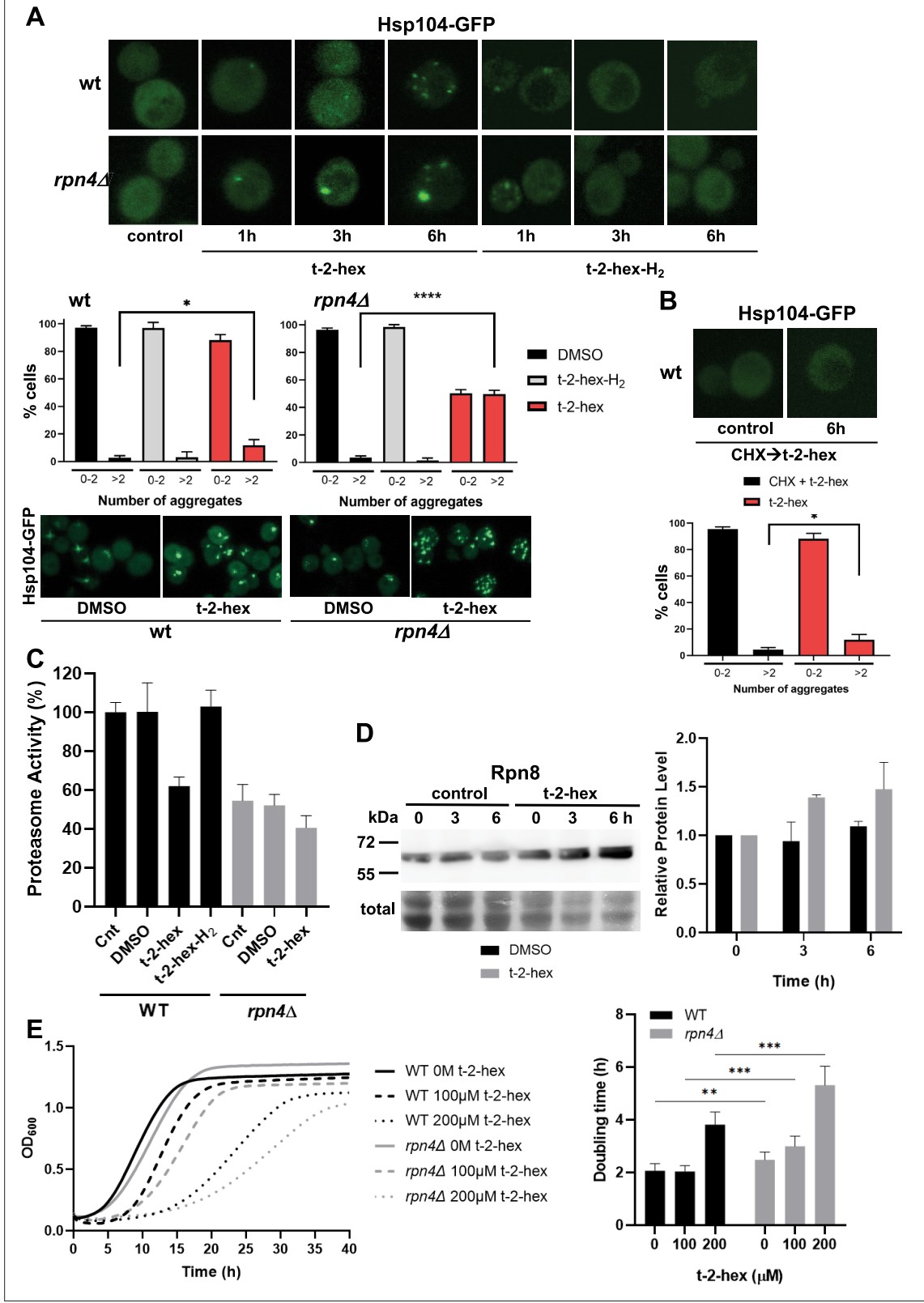

**Figure 4.** Trans-2-hexadecenal (t-2-hex) leads to cytosolic protein aggregation and inhibition of proteasomal function. (**A**) GFP-tagged Hsp104 was used to visualize intracellular protein aggregation in wild-type and proteasome-deficient *rpn4Δ* cells. Specifically, unsaturated t-2-hex caused a slowly increasing protein aggregation. Cells were treated with 100 µM of the bioactive lipids for the indicated times. Lower panels: Quantification of t-2-hex-induced protein aggregates across cell populations. Number of analyzed cells: wt DMSO n=601, wt t-2-hex n=584, wt t-2-hex-H₂ n=558; rpn4

*Figure 4 continued on next page*

*Figure 4 continued*

DMSO n=551, rpn4 t-2-hex n=549, rpn4 t-2-hex-H$_2$ n=329. (**B**) t-2-hex activated protein aggregation was no longer observed after inhibition of protein synthesis with cycloheximide (CHX). Lower panel: Quantification of t-2-hex induced protein aggregates across cell populations. Number of analyzed cells: t-2-hex n=584, CHX +t-2-hex n=134. (**C**) Effect of pro-apoptotic t-2-hex on proteasomal activity. Proteasomal activity was quantified in whole cell extracts before (Cnt) or after treatment with 200 µM t-2-hex, t-2-hex-H$_2$, or vehicle (DMSO) for 3 hr in wild-type or *rpn4Δ* cells (n=3). Activity of untreated wild-type cells was set to 100%. (**D**) Response of proteasomal subunit Rpn8 expression upon t-2-hex exposure. Rpn8 was expressed as a Tap fusion from its chromosomal locus and cells were treated or not with 200 µM t-2-hex for the indicated times. Rpn8 protein abundance was quantified by anti-Tap western blot (upper panel) and quantified relative to uninduced levels (n=2). (**E**) Proteasomal deficiency causes t-2-hex sensitivity. Growth of yeast wild-type and *rpn4Δ* cells was quantified upon the indicated t-2-hex concentrations (left panel) and the doubling time calculated (right panel), n=6. *p<0.05, **p<0.01, ***p<0.001, ****p<0.0005 by Student´s unpaired t-test.

The online version of this article includes the following source data for figure 4:

**Source data 1.** PDF file containing the original uncropped western blots for *Figure 4D*, indicating the relevant bands.

**Source data 2.** Original files for western blot analysis displayed in *Figure 4D*.

impaired the import of mitochondrial precursor proteins. A defect in the import of mitochondrial proteins with a cleavable signal peptide can be visualized by the immunological detection of the larger, unprocessed pre-protein. We chose three different mitochondrial proteins for this survey: Aim17, whose expression is strongly activated upon t-2-hex stress (42 fold according to our RNA-seq study), the Cox5a subunit of CIV at the inner mitochondrial membrane and the Ilv6 regulator of isoleucine and valine biosynthesis in the mitochondrial matrix. We found for all three proteins that t-2-hex treatment of wild-type cells led to the rapid appearance of the corresponding pre-proteins, indicating an efficient block of mitochondrial protein import by the lipid (*Figure 5A*). The Aim17 and Cox5a pre-proteins correlated with the same precursors induced by the general mitochondrial uncoupler CCCP. However, in the Ilv6 case, we observed the appearance of yet an additional precursor form specific for t-2-hex. The behavior of Aim17 is especially interesting, because this protein is massively made de novo in response to t-2-hex exposure. We observe that its pre-protein quickly accumulates to very high levels, which equal the abundance of the mature Aim17 protein. This strongly suggests that t-2-hex very efficiently blocks mitochondrial pre-protein transport and processing, and that most of the newly synthesized mitochondrial precursors are not correctly introduced into mitochondria upon pro-apoptotic lipid stress. We next confirmed that this inhibition was specific for the unsaturated t-2-hex form, as we did not detect precursor accumulation of four different mitochondrially imported proteins by the saturated t-2-hex-H$_2$ (*Figure 5B*). We additionally determined general mitochondrial fragmentation, which had been reported previously for t-2-hex (*Manzanares-Estreder et al., 2017*). Also in this case, we found that mitochondrial fragmentation was specific for the unsaturated lipid aldehyde (*Figure 5C*), indicating that mitochondrial fragmentation and inhibition of protein import depended on the protein lipidation capacity of t-2-hex.

The observation that newly made Aim17 was not processed properly and accumulated in large amounts during the first hour of t-2-hex stress, prompted us to investigate the long-term effects of t-2-hex on proteostasis. For these studies, we initially selected two more highly inducible mitochondrial proteins apart from Aim17, the cytochrome c variant Cyc7 targeted to the mitochondrial intermembrane space and the pyruvate carrier subunit Mpc3 targeted to the inner mitochondrial membrane. All proteins were visualized in yeast wild-type cells, which were treated with t-2-hex for up to 6 hr. As expected, we found that the Aim17 pre-protein also accumulated at prolonged t-2-hex exposure (*Figure 5D*). However, we additionally were able to see a smear of aberrant, SDS-resistant protein forms in the larger molecular weight range, which could be an indication of aggregated Aim17. Aberrant protein forms were also detected for Cyc7 and, in a very abundant manner, for Mpc3, which both lack a defined cleavable signal peptide. Although the expression of all three mitochondrial proteins was activated, the amount of the mature protein forms did not increase (*Figure 5D*). This suggested that the newly synthesized mitochondrial proteins never get properly incorporated in the organelle upon t-2-hex stress and instead remain in the cytoplasm, where they increasingly misfold and aggregate. Indeed, cycloheximide treatment completely abolished the formation of Mpc3 and Aim17 aggregates (*Figure 5E*), indicating that t-2-hex only affects the proteostasis of de novo synthesized mitochondrial precursor proteins.

Clogging the mitochondrial import machinery has been shown to cause a generalized proteostatic problem in the cytoplasm, most likely by saturating the cellular chaperone system (*Boos et al.,*

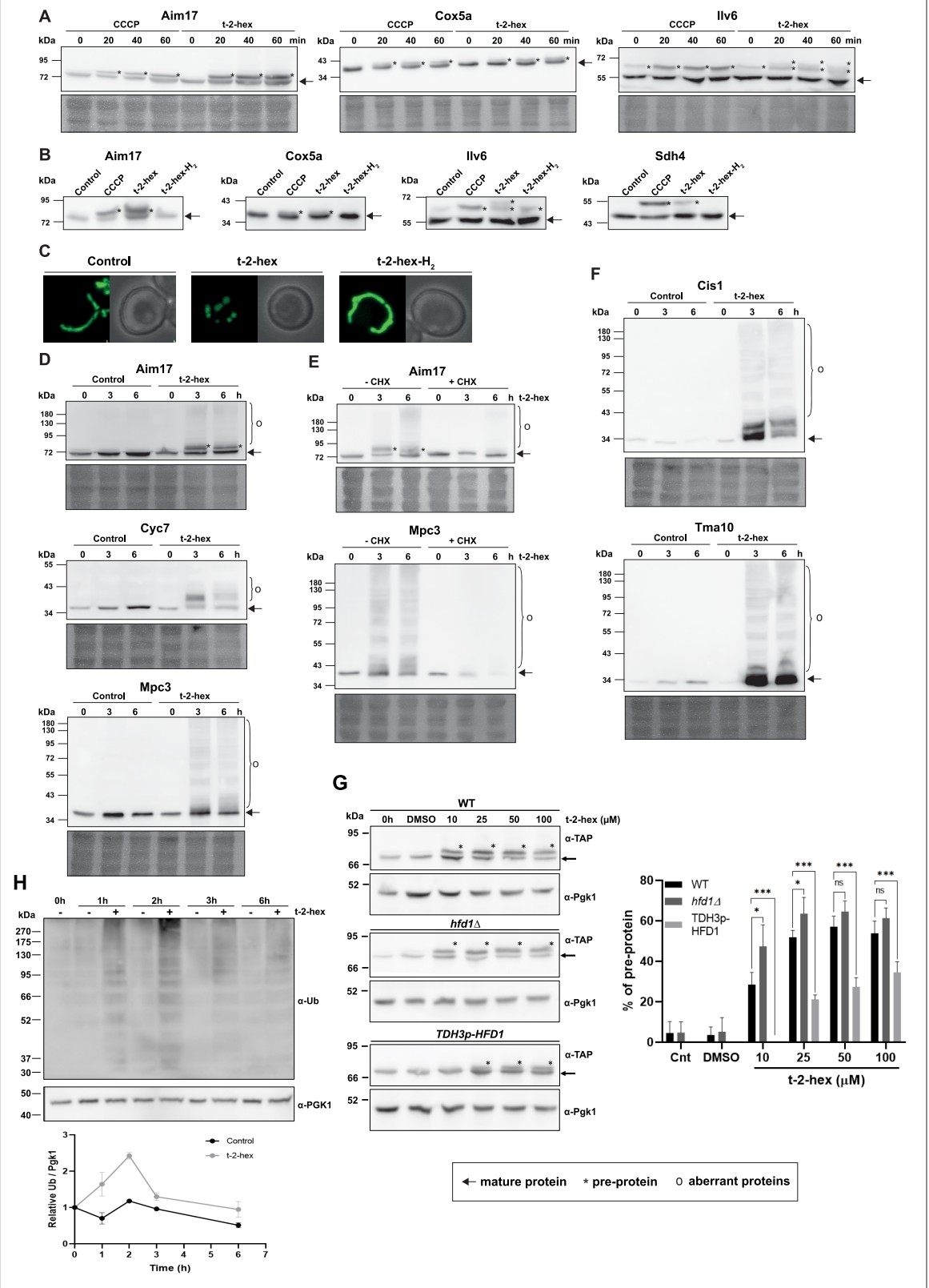

**Figure 5.** Trans-2-hexadecenal (t-2-hex) overload induces mitochondrial pre-protein accumulation and aggregation of de novo synthesized proteins. (**A**) The appearance of unimported mitochondrial precursor proteins (*) was induced by the uncoupler CCCP (20 μM) or t-2-hex (200 μM) for the indicated times. Aim17, Cox5a, Ilv6, and Sdh4 were visualized in chromosomally Tap-tagged wild-type yeast strains by anti-Tap western blot. (**B**) Mitochondrial import block depends on t-2-hex unsaturation. The same strains as in (**A**) were treated with DMSO (control), CCCP (20 μM), t-2-hex

*Figure 5 continued on next page*

*Figure 5 continued*

(200 µM), or t-2-hex-H$_2$ (200 µM) for 40 min and mitochondrial pre-protein accumulation was visualized by anti-Tap western blot. (**C**) Mitochondrial fragmentation depends on t-2-hex unsaturation. Yeast wild-type cells expressing mt-GFP were treated with vehicle (control), 200 µM t-2-hex or t-2-hex-H$_2$ for 1 hr. (**D**) t-2-hex induces the formation of aberrant mitochondrial proteins. Yeast wild-type strains expressing Aim17-, Cyc7-, or Mpc3-Tap tagged fusion proteins from their chromosomal locus were treated with 200 µM t-2-hex for the indicated times. Fusion proteins were detected by anti-Tap western blot. (**E**) Inhibition of de novo protein synthesis abolishes the formation of aberrant mitochondrial proteins. Experimental conditions as in (**D**), but including a pretreatment with cycloheximide (CHX, 250 µg/ml) where indicated. (**F**) t-2-hex induces aberrant forms of highly expressed proteins. Yeast wild-type strains expressing Cis1- or Tma10-Tap tagged fusion proteins from their chromosomal locus were treated or not with 200 µM t-2-hex for the indicated times. Fusion proteins were detected by anti-Tap western blot. (**G**) Hfd1 gene dose determines the extent of t-2-hex-induced pre-protein accumulation. Aim17 pre-protein accumulation in response to different t-2-hex concentrations was quantified in wild-type, *hfd1Δ* and *HFD1* overexpressing cells. Cells were either untreated, or treated with the indicated lipid concentrations or DMSO for 20 min. The percentage of pre-protein relative to total Aim17 protein was calculated in the right panel (n=4) (**H**) t-2-hex causes rapid protein ubiquitination. Yeast wild-type cells were treated for the indicated times with t-2-hex (200 µM) and protein ubiquitination visualized by anti-Ub western blot (upper panel) and quantified relative to the Pgk1 loading control (lower panel) (n=2).

The online version of this article includes the following source data and figure supplement(s) for figure 5:

**Source data 1.** PDF file containing the original uncropped western blots for *Figure 5*, indicating the relevant bands.

**Source data 2.** Original files for western blot analysis displayed in *Figure 5A-D*.

**Source data 3.** Original files for western blot analysis displayed in *Figure 5E-H*.

**Figure supplement 1.** Quantification of the inhibition of mitochondrial protein import by cytometric mt-GFP measurements.

*2019Wrobel et al., 2015*). We, therefore, looked at other t-2-hex inducible proteins, which are not transported through the mitochondrial membranes, Cis1 and Tma10. The MitoCPR protein Cis1 is induced upon mitochondrial protein import stress and attaches to the outer mitochondrial transport complex without being imported. Tma10 is the most up-regulated protein upon t-2-hex stress according to our RNA-Seq analysis and localizes to the cytoplasm. For both proteins, we confirm a strong increase in abundance upon t-2-hex treatment (*Figure 5F*). More importantly, also Cis1 and Tma10 show an abundant aggregation when they are massively synthesized in the presence of the pro-apoptotic lipid (*Figure 5F*), suggesting that t-2-hex probably affects correct folding of a large number of newly synthesized mitochondrial and cytosolic proteins. We next tested how sensitively t-2-hex interfered with mitochondrial protein import by applying low lipid doses. As shown in *Figure 5G*, Aim17 pre-protein accumulated by the external addition of just 10 µM of t-2-hex. This inhibition was aggravated in a *hfd1Δ* mutant and alleviated by *HFD1* overexpression (*Figure 5G*). These data indicated that mitochondrial pre-protein accumulation was induced by physiological t-2-hex concentrations and that its extent was regulated by the activity of detoxifying enzymes such as Hfd1. Misfolded cytosolic proteins are rapidly marked by ubiquitination for their proteolytic degradation by the proteasome. We, therefore, tested whether t-2-hex overload increased overall protein ubiquitination. We performed anti-Ub western blot analyses in a time course of t-2-hex treatment in whole cell extracts. As shown in *Figure 5H*, t-2-hex induced general protein ubiquitination in a rapid and transient manner.

We next examined the inhibition of mitochondrial protein import by t-2-hex by flow cytometric GFP assays in live yeast cells (*Sirk et al., 2003*). We compared the induced expression and maturation of GFP, either targeted to the cytosol or to the mitochondrial matrix (*Figure 5—figure supplement 1A*). We found that low t-2-hex doses (≥5 µM) significantly inhibited mitochondrial GFP (mt-GFP) activity as compared to cytosolic GFP in wild-type cells (*Figure 5—figure supplement 1B*). This inhibition was not observed with the saturated t-2-hex-H$_2$. We further examined whether t-2-hex could modulate mitochondrial protein import indirectly by interfering with the mitochondrial membrane potential (ΔΨm). Cytometric rhodamine123 assays revealed that t-2-hex did not alter ΔΨm in the range of concentrations which efficiently inhibited mt-GFP activity (*Figure 5—figure supplement 1C*). We additionally tested whether the t-2-hex induced decrease in proteasome activity could indirectly reduce mitochondrial protein import. This was not the case, as *rpn4* mutant cells exhibited a normal mt-GFP activity (*Figure 5—figure supplement 1D*). These data suggested that t-2-hex sensitively inhibited mitochondrial pre-protein import.

## A functional genomics approach reveals pro- and anti-apoptotic gene functions in response to t-2-hex

In order to identify genetic determinants, which would modulate pro-apoptotic t-2-hex toxicity, we performed a saturated transposon mutagenesis screen (SATAY; *Michel et al., 2017*). We generated a yeast mutant library consisting of several millions of independent transposon insertion mutants in an *hfd1Δ* background (*Figure 6A*). The pooled library was then grown in the presence or absence of growth-limiting amounts of t-2-hex. The transposon insertion sites (TNs) were mapped along the entire yeast genome by massive sequencing. The total number of transposon reads or the number of different TNs per gene was then used as an indicator of the fitness of all mutants upon pro-apoptotic lipid stress conditions. In this scenario, under-represented genes (statistically lower number of TNs in a given gene in the t-2-hex versus the untreated library) have potentially anti-apoptotic functions against the bioactive lipid (*Supplementary file 4*), while over-represented genes (statistically higher number of TNs in a given gene) have potentially pro-apoptotic functions supporting the toxicity of t-2-hex (*Supplementary file 5*). We analyzed the potential anti-apoptotic SATAY hits with a $\log_2 <-0.75$ according to expected detoxification pathways (heat shock response, pleiotropic drug response), to their function in the ER (the intracellular site where t-2-hex is generated) or in mitochondria (the major t-2-hex target identified so far). As depicted in *Figure 6B*, transposon mutations in the *HSF1* gene encoding heat shock factor are strongly counter-selected by t-2-hex stress. *HSF1* is essential in yeast; however, mutations in the Hsf1 C-terminal region are viable and reduce the activity of the transcription factor (*Chen et al., 1993*). Untreated cells accumulate TN insertions in this region, however, the Hsf1 C-terminus becomes essential upon t-2-hex treatment, thus demonstrating that the heat shock response is rate limiting during the cellular defense against the lipid. Additionally, the pleiotropic drug response (PDR) is identified here as an important layer of defense against t-2-hex. Both the dominant PDR transcriptional activator Pdr1 and the Snq2 PDR extrusion pump of the ABC type are among the gene functions, which are not allowed to accumulate TN insertions upon t-2-hex overload. Furthermore, we identify several mitochondrial functions (*FMP52, CYC3, HEM14, MRP7*), which become important upon apoptotic lipid stress. They play essential functions in heme biosynthesis (Hem14) and cytochrome c maturation (Cyc3), or in mitochondrial protein synthesis (Mrp7). Of note, Fmp52 is a member of the short-chain dehydrogenase/reductase (SDR) family, which suggests a function in the enzymatic detoxification of t-2-hex in addition to Hfd1. We furthermore identify a group of ER-resident proteins as crucial for t-2-hex tolerance. Apart from the ER membrane proteins Agp2, Nnf2, Cwh43, and Spf1, we found two proteins, Eps1 and Usa1, specifically involved in the ER-associated protein degradation pathway (ERAD) as important for t-2-hex tolerance.

Unexpectedly, the SATAY screen revealed a large number of genes, which upon TN mutation, partially suppressed sensitivity to t-2-hex stress (*Figure 6B*). A closer inspection of these gene functions with a $\log_2 >1.5$ revealed that most of them belonged to the functional groups 'Cytosolic ribosome and translation' and 'Amino acid metabolism.' Related to cytosolic protein biosynthesis, we identified proteins involved in ribosomal RNA processing (*DIM1, RIO2, ROK1, RRP3, RRP15, SRM1*), ribosome biogenesis (*MRT4, NOP8, BCP1, TMA23, UTP25*), aa-tRNA synthesis (*ALA1, CRS1, FRS1, ILS1, TRM9*), pre-RNA processing (*PRP5*), translation (*CDC123, NIP1, NAM7*), and ribosomal protein subunits (*RPL6A, RPS1A*+12 others). Related to amino acid metabolism, enzymes providing precursors for general amino acid biosynthesis (*ARO10, DFR1, SHM2*), as well as regulators and enzymes of the synthesis of specific amino acids (*ARG7, ARG82, ASP1, GFA1, ILV3, LYS12, LYS14, ORT1, SER2*) were identified in this group of potential pro-apoptotic functions. Taken together, these results suggested that an efficient heat shock and PDR response, as well as specific mitochondrial and ER functions were required, while a high rate of cytosolic protein biogenesis was detrimental, for the survival upon apoptotic lipid stress.

Indeed, we confirmed that loss of function of the PDR transcriptional activator Pdr1 caused a moderate, but consistent growth delay in the presence of t-2-hex (*Figure 7A*). Interestingly, we found that deletion of mitochondrial Fmp52 function caused a severe sensitivity to t-2-hex exposure (*Figure 7A*). Fmp52 belongs structurally to the NAD-binding short-chain dehydrogenase/reductase (SDR) family and localizes to highly purified mitochondrial outer membranes (*Zahedi et al., 2006*). This might suggest that Fmp52 participates in the enzymatic detoxification of t-2-hex in addition to Hfd1.

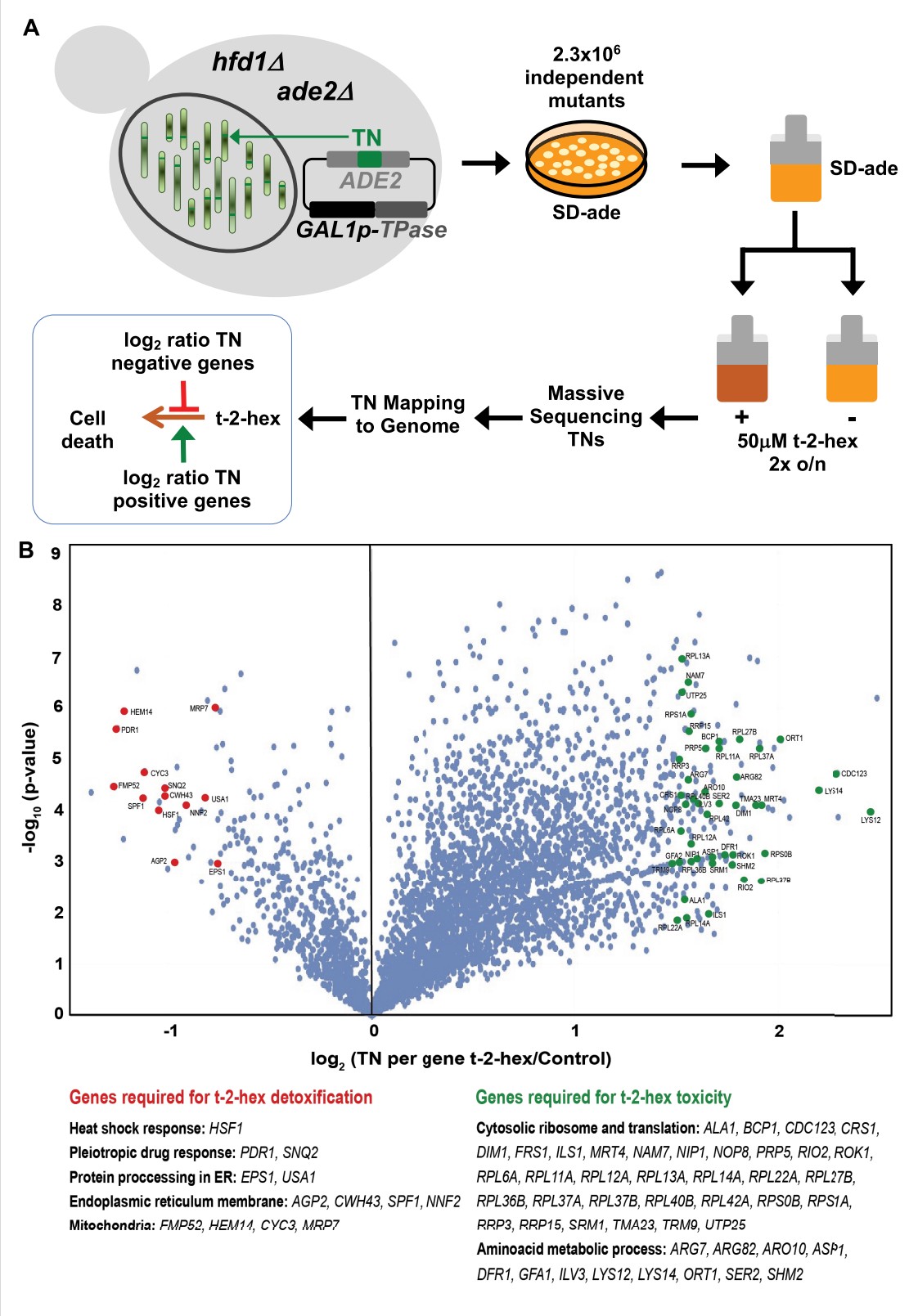

**Figure 6.** Functional genomics screen SATAY for the identification of pro- and anti-apoptotic functions upon trans-2-hexadecenal (t-2-hex) stress. (**A**) Schematic outline of the SATAY experiment. *hfd1Δ* cells harboring the galactose-inducible transposase (TPase) and a transposon (TN) disrupting the *ADE2* gene were grown in galactose-containing medium to induce transposition. TN-generated mutant cells were inoculated in synthetic glucose medium lacking adenine in the presence or absence of 50 µM t-2-hex, grown for several generations, and harvested for DNA extraction and sequencing

*Figure 6 continued on next page*

*Figure 6 continued*

of transposon insertion sites (TNs). TNs are mapped to the genome to identify genes that become required for proliferation under t-2-hex stress conditions (red) or genes whose mutation is beneficial for t-2-hex tolerance (green). (**B**) Identification of anti- and pro-apoptotic gene functions upon t-2-hex overload. Volcano plot showing the fold change of number of transposon insertions (TN) per gene of libraries grown in t-2-hex excess versus control conditions. TN under-enriched (anti-apoptotic) genes were analyzed with a $\log_2$ ratio below –0.75 and are summarized in *Supplementary file 4*. TN-enriched (pro-apoptotic) genes were analyzed with a $\log_2$ ratio >1.5 and are available in *Supplementary file 5*.

We next wanted to know whether a general connection existed between the de novo protein synthesis rate and the susceptibility to pro-apoptotic t-2-hex. As a first approach, we tested several deletion mutants in specific subunits of the small and large ribosomal subunits from the yeast knockout collection for their individual t-2-hex tolerance. As it is known that ribosomal subunit mutants have divergent effects on protein synthesis rates and rapidly accumulate compensatory mutations (*Steffen et al., 2008*), we included several mutant strains with diverse impacts on overall growth rates in this study. As shown in *Figure 7B*, the *rps30b* mutant had the greatest reduction in proliferation rate and showed the highest tolerance to t-2-hex. The *rpl16a* mutant with a minimal growth reduction was only slightly resistant to the apoptotic lipid, while other ribosomal subunit mutants (*rps28a*, *rpl40a*) with unaffected growth rates were t-2-hex sensitive comparable to the wild-type. This indicated that the loss of particular ribosomal subunits was able to improve t-2-hex tolerance in cases where growth rate and presumably protein synthesis rates were decreased. We then tested this hypothesis directly by the application of the ribosomal inhibitor diazaborine (DAB), which specifically interferes with ribosomal biogenesis and protein synthesis in yeast (*Pertschy et al., 2004*). As shown in *Figure 7C*, growth inhibitory DAB concentrations consistently made yeast cells more tolerant to t-2-hex, indicating that a slow cytosolic protein synthesis was beneficial to counteract the apoptotic lipid. Finally, we applied an independent intervention to slow down protein synthesis, which is the switch from glucose to galactose growth. Metabolic flux through the galactose pathway is reduced by >30% as compared to glucose, which slows down cell proliferation (*Ostergaard et al., 2000*). We confirmed that galactose growth improved the resistance of yeast cells to t-2-hex inhibition, both in the wild-type and the *hfd1Δ* background (*Figure 7D*).

We next compared the pro- and anti-apoptotic gene functions indicated by the SATAY assay with the previously generated RNAseq profiles. We found that anti-apoptotic functions such as heat shock factor, pleiotropic drug response, and specific mitochondrial proteins showed consistently positive transcriptional regulation (*Figure 7E*, red), while the expression of the majority of pro-apoptotic functions involved in cytosolic protein synthesis and general amino acid metabolism was down-regulated (*Figure 7E*, green).

## Apoptotic lipid signaling via t-2-hex and its detoxification take place at the mitochondrial TOM complex

Having found that the pro-apoptotic lipid t-2-hex impaired mitochondrial protein import and caused a general proteostatic readjustment in the cell, we next wanted to know whether t-2-hex and its detoxifier Hfd1 target the mitochondrial pre-protein import machinery. The essential import pore at the outer mitochondrial membrane is formed by the TOM complex. The yeast core TOM complex contains the Tom40, Tom22, Tom5, Tom6, and Tom7 subunits, which associate with the Tom70 and Tom20 receptor proteins to form the functional translocase complex (*Araiso et al., 2022*). We first focused on the interaction of Hfd1 with the TOM complex. It had been previously shown that the Hfd1 lipid aldehyde dehydrogenase co-purified with Tom22 (*Opaliński et al., 2018*) and Tom70 (*Burri et al., 2006*) in high throughput studies. Additionally, Hfd1 seems to co-localize with a Tom70-containing complex in a recent high-resolution complexome profiling study (*Schulte et al., 2023*). We, therefore, studied the interaction of Hfd1 with Tom22 and Tom70 in targeted pulldown experiments from whole cell extracts. As shown in *Figure 8A*, we found that Hfd1 co-purified weakly with Tom22, however, it was more efficiently pulled down with Tom70. These data suggested that at least a portion of Hfd1 is physically associated with the TOM mitochondrial protein import machine with Tom70 as a potential interactor. These data indicated that t-2-hex might act directly at the TOM complex. We, therefore, proceeded with biochemical assays to elucidate a possible direct modification of TOM subunits by the apoptotic lipid.

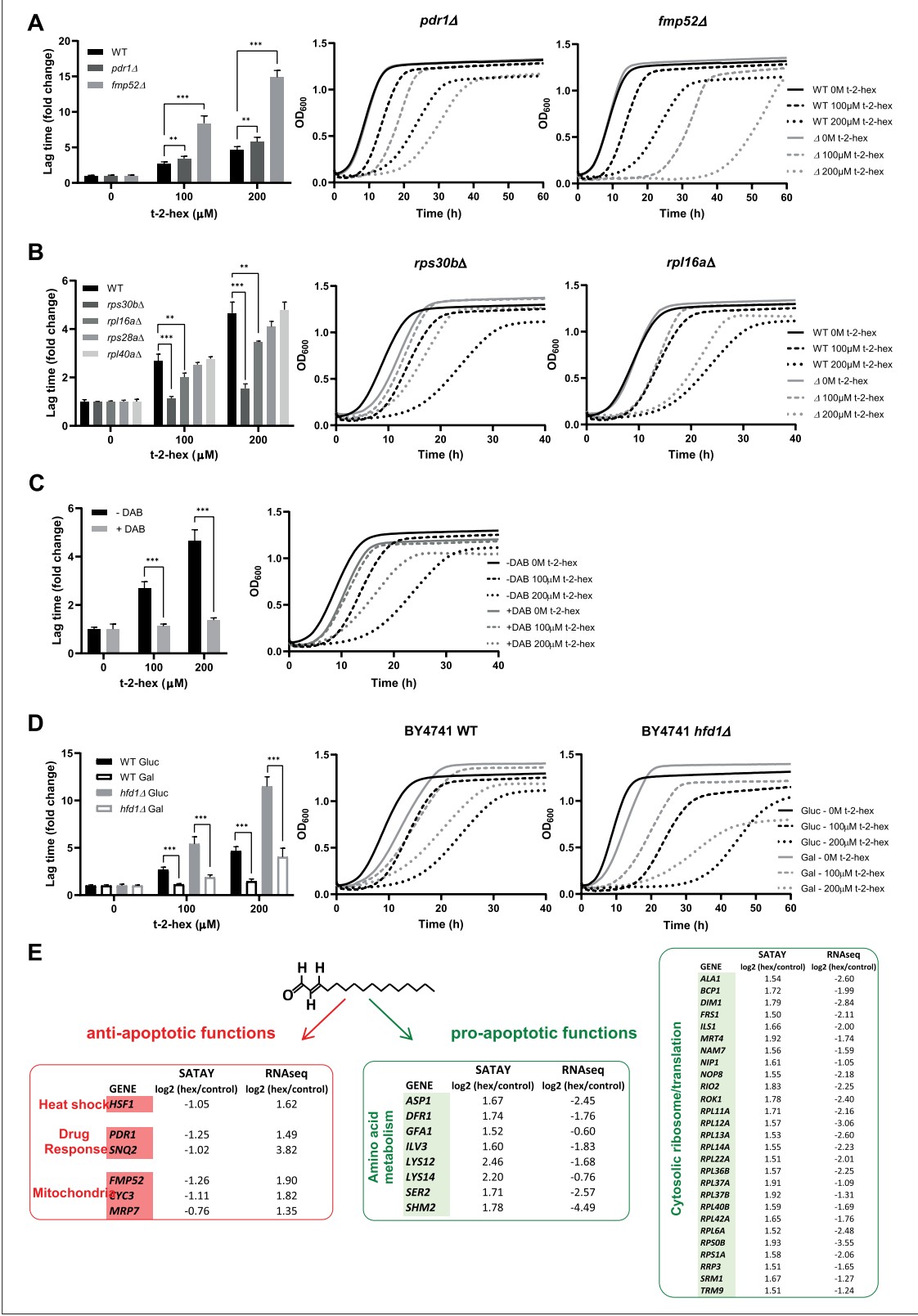

**Figure 7.** Functional analysis of pro- and anti-apoptotic genes upon trans-2-hexadecenal (t-2-hex) excess. (**A**) The pleiotropic drug response activator Pdr1 and the mitochondrial protein Fmp52 are necessary for t-2-hex tolerance. Quantitative growth assay of wild-type, *pdr1Δ*, and *fmp52Δ* cells upon the indicated concentrations of t-2-hex. The relative lag time was quantified as an estimator of growth inhibition (n=3). (**B**) Deletion of ribosomal protein subunits causes t-2-hex resistance. As in (**A**), but using the yeast deletion strains *rps30bΔ*, *rpl6aΔ*, *rps28aΔ*, and *rpl40aΔ*. Growth curves are depicted

*Figure 7 continued on next page*

*Figure 7 continued*

only for the t-2-hex tolerant strains *rps30bΔ* and *rpl6aΔ* (n=3). (**C**) Inhibition of ribosomal biogenesis is beneficial for t-2-hex tolerance. Yeast wild-type cells were grown upon the indicated t-2-hex concentrations in the presence or absence of the inhibitor diazaborine (DAB, 20 µg/ml) (n=3). (**D**) Galactose growth improves t-2-hex tolerance. Growth of wild-type and *hfd1Δ* cells upon the indicated concentrations of t-2-hex on synthetic glucose or galactose media (n=3). **p<0.01, ***p<0.001 by Student´s unpaired t-test. (**E**) Summary of anti-apoptotic genes identified by SATAY with significant transcriptional up-regulation (red) and pro-apoptotic genes identified by SATAY with significant transcriptional down-regulation (green) according to our RNA-seq study.

We performed in vitro chemoproteomic experiments using the clickable t-2-hex analogue t-2-hex-alkyne on enriched intact mitochondrial fractions of yeast wild-type cells, applying both high (100 µM) and low (10 µM) lipid doses (*Figure 8B*). After alkylation of mitochondrial proteins and the subsequent addition of biotin, we identified the t-2-hex targets by mass spectrometry. We found that several subunits of the TOM complex, including the Tom40 core channel and the accessory Tom70 and Tom71 proteins, were significantly enriched in the t-2-hex treated samples (*Figure 8B*, *Supplementary file 6*). We additionally identified with less intensity the small Tom5 and Tom7 subunits, which lack cysteine residues and thus are most likely pulled down in these assays because of their physical association with truly lipidated Tom subunits. Very interestingly, we additionally identified several essential subunits of the Tim23 inner mitochondrial membrane protein import complex as t-2-hex targets (*Figure 8B*). This indicates that apart from the TOM complex, the pro-apoptotic lipid directly modifies subunits of the Tim23 protein translocator. Of the core TOM complex from yeast, only the Tom40 channel contains cysteine residues and interestingly has been identified as a t-2-hex target in a chemoproteomic screen on whole human cell extracts (*Jarugumilli et al., 2018*). We next tested for direct lipidation of Tom40 in mitochondrial fractions of yeast cells with physiological levels of HA-tagged Tom40. Low micromolar concentrations (25 µM) of the t-2-hex probe were sufficient to specifically modify Tom40 in these lipidation assays (*Figure 8C*), showing that the Tom40 protein is a direct lipidation target in vitro. Competition assays with free thiol groups confirmed that t-2-hex likely adds to cysteine residues in the Tom40 channel (*Figure 8C*). We next addressed the question of whether individual TOM subunits were important to mediate t-2-hex toxicity. We first characterized a mutant, which expressed a cysteine-free version of Tom40 (Tom40CFREE), thus impairing cysteine lipidation at Tom40. As shown in *Figure 8D*, the Tom40CREE strain showed undistinguishable tolerance to t-2-hex as compared to wild-type, suggesting that Tom40 lipidation alone did not explain the growth arrest of the pro-apoptotic lipid. We, therefore, extended our study to other, non-essential accessory TOM subunits Tom20 and Tom70. In this case (*Figure 8D*), we observed a significant increase in t-2-hex tolerance caused by the deletion of either *TOM20* or *TOM70*. We further confirmed that a *tom20* mutant strain accumulated less Aim17 pre-protein upon t-2-hex stress (*Figure 8—figure supplement 1*). These results identify the mitochondrial protein import machinery as a physiologically important target for the pro-apoptotic lipid t-2-hex, which is mediated by several Tom subunits and not just the Tom40 import channel. In context with all previous results of our study, we propose a model where pro-apoptotic lipid signaling and cell proteostasis are functionally linked via mitochondrial protein import (*Figure 9*). Upon excess of t-2-hex or lack of Hfd1 function, lipidation of the TOM complex and subsequent failure of mitochondrial precursor import causes a general proteostatic imbalance. If this imbalance persists it can engage the cell in apoptotic cell death.

## Discussion

Maintaining proteostasis is essential for cell survival and is tightly regulated to prevent the accumulation of misfolded or aggregated proteins. Proteostatic imbalance is known to be a key driver for the induction of apoptotic cell death and cellular aging (*Kaushik and Cuervo, 2015*). Although it has been well established that many extracellular stresses trigger the entry into apoptotic self-destruction programs and many executors of these apoptotic programs have been revealed (*Kist and Vucic, 2021*), it remains largely unknown how intracellular signals perturb mitochondrial function and proteostasis in order to push the balance towards a regulated cell death. Our present work reveals a novel pro-apoptotic mechanism, which consists of the direct inhibition of mitochondrial protein import by bioactive lipids generated from the sphingolipid degradation pathway, thereby directly linking pro-apoptotic lipids with mitochondrial and general protein homeostasis and aggregation.

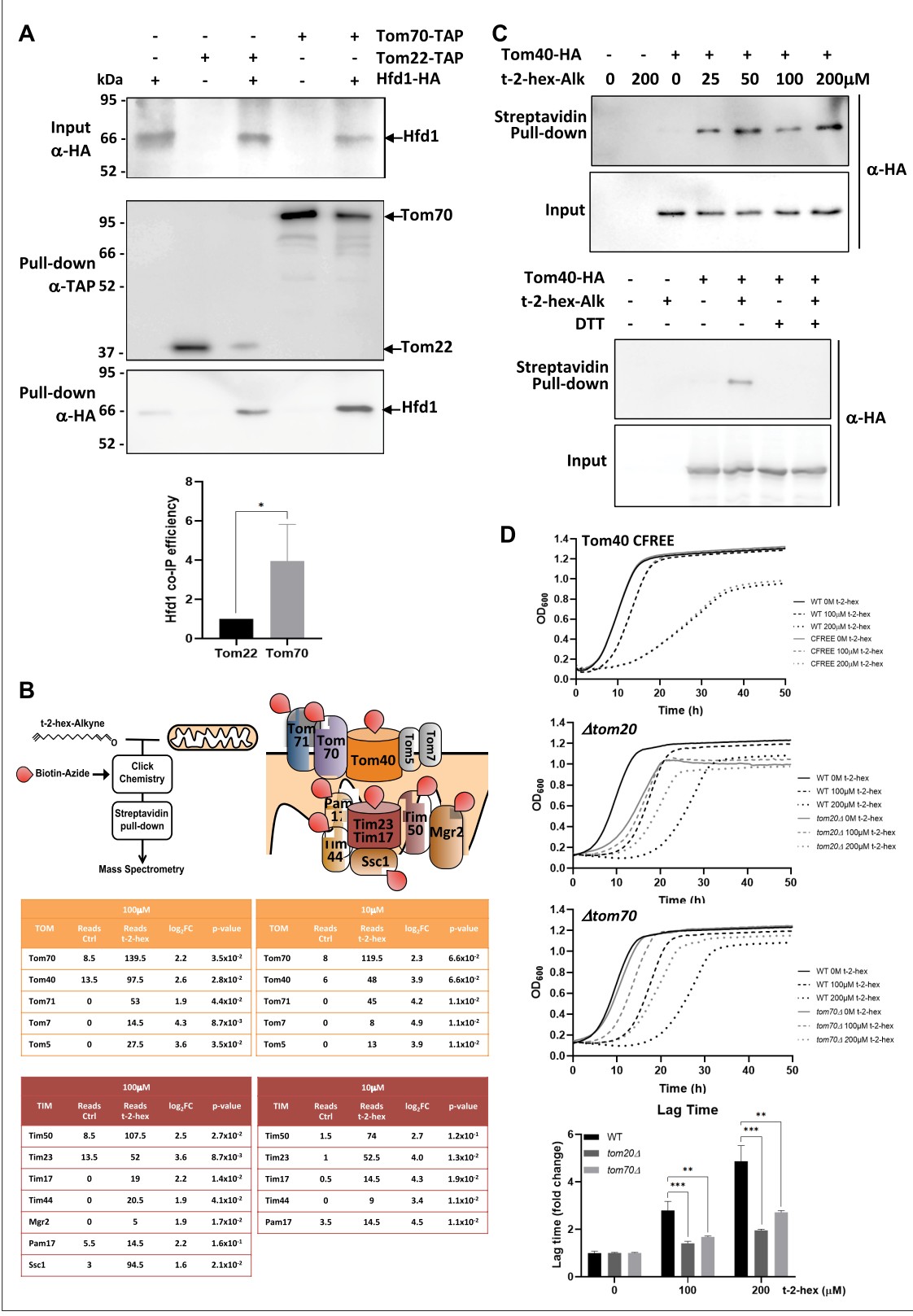

**Figure 8.** t-2-hex targets the mitochondrial TOM complex. (**A**) Hfd1 co-purifies with Tom70. Constitutively Hfd1-HA expressing cells were used in co-immunoprecipitation experiments from yeast whole cell extracts in the presence or not of endogenously expressed Tom22- or Tom70-Tap. Lower panel: Quantification of Hfd1 co-purification with respect to Tom20 and Tom70; n=6. (**B**) A chemoproteomic screen with t-2-hex alkyne identifies the TOM and Tim23 complexes as lipidation targets. Purified mitochondrial fractions from yeast wild-type cells were treated or not with the clickable

*Figure 8 continued on next page*

*Figure 8 continued*

analogue t-2-hex-Alkyne at high and low doses (100 µM or 10 µM). After addition of biotin to the modified proteins and purification with Streptavidin agarose pull-down, the protein identities of the trans-2-hexadecenal (t-2-hex) targets were determined by mass spectrometry. Cysteine-containing subunits of the TOM and Tim23 complexes are depicted as direct lipidation targets. The tables show the mean spectral reads for selected subunits in the chemoproteomic analysis for mock-treated (Ctrl) and t-2-hex treated mitochondria (n=2), as well as the $\log_2$ fold enrichment and adjusted p-value. (**C**) Tom40 is lipidated in vitro by t-2-hex. Upper panel: t-2-hex-Alkyne was used in the indicated concentrations to lipidate proteins in enriched mitochondrial preparations from Tom40-HA expressing cells or control cells. After t-2-hex addition, input samples were generated directly, while pull-down samples were treated with click chemistry for covalent biotin linkage and subsequent Streptavidin purification. Tom40-HA was detected in all samples by anti-HA western blot. Lower panel: Competition of Tom40 t-2-hex lipidation by free thiol groups. t-2-hex-Alkyne was used at 100 µM to lipidate Tom40-HA in mitochondrial preparations in the presence or absence of 2 mM DTT. (**D**) Mutations of TOM accessory subunits 20 or 70 cause t-2-hex tolerance. t-2-hex toxicity assays comparing wild-type cells with the cysteine-free Tom40 mutant (Tom40 CFREE) and deletion mutants *tom20* or *tom70*. Lower panel: Quantitative comparison of growth performance (length of lag phase) of the *tom20* and *tom70* strains upon the indicated t-2-hex concentrations (n=3). \*\*p<0.01, \*\*\*p<0.001 by Student´s unpaired t-test.

The online version of this article includes the following source data and figure supplement(s) for figure 8:

**Source data 1.** PDF file containing the original uncropped western blots for ***Figure 8A and C***, indicating the relevant bands.

**Source data 2.** Original files for western blot analysis displayed in ***Figure 8A and C***.

**Figure supplement 1.** Tom20 function determines the extent of trans-2-hexadecenal (t-2-hex)-induced pre-protein accumulation.

**Figure supplement 1—source data 1.** PDF file containing the original uncropped western blots for ***Figure 8—figure supplement 1***, indicating the relevant bands.

**Figure supplement 1—source data 2.** Original files for western blot analysis displayed in ***Figure 8—figure supplement 1***.

It was a groundbreaking discovery that bioactive lipids such as t-2-hex in ER microsomes were in great part necessary and responsible for MOMP and mitochondrial cell death in human cells (***Chipuk et al., 2021***; ***Chipuk et al., 2012***; ***Renault and Chipuk, 2013***). The recent finding that t-2-hex is a direct pro-apoptotic modulator of a single cysteine residue in human Bax provided an explicit molecular mechanism of MOMP induction (***Cohen et al., 2020***; ***Jarugumilli et al., 2018***). However, t-2-hex belongs to a group of lipid-derived electrophiles (LDEs), which are capable of directly modifying free cysteines in target proteins in a non-enzymatic manner and have been suggested to act as endogenous messengers to modulate signal transduction pathways within the cell (***Fritz and Petersen, 2013***; ***Sousa et al., 2017***). This implies that t-2-hex has potentially many protein targets in the cell, which has been further demonstrated by in vitro chemoproteomic experiments in human cells, revealing several hundred alkylation targets including Tom40 (***Jarugumilli et al., 2018***). It remains challenging and important to identify the biologically relevant protein targets of LDEs and especially t-2-hex (***Chen et al., 2016***; ***Wang et al., 2014***). Our combination of in vitro lipidation assays and functional genomics surveys identifies the mitochondrial outer membrane protein transport machine as a selective target for apoptotic signaling mediated through t-2-hex. Moreover, the lipid-mediated inhibition of the mitochondrial TOM complex, here detailed in the yeast model, provides a general apoptotic mechanism, which might have been established early in evolution, given the conservation of the core subunits Tom40 or Tom70 from fungi to man (***Mani et al., 2016***).

Mitochondrial protein import might be a strategic target for apoptotic signaling, because the import of more than a thousand cytoplasmatically synthesized pre-proteins into mitochondria is a major traffic hub in eukaryotic cells, which is under exquisite surveillance to avoid clogging of the organelle (***Song et al., 2021***). The aberrant accumulation of mitochondrial precursor proteins in the cytosol leads to proteotoxic stress, which negatively affects cell viability (***Nargund et al., 2012***; ***Wang and Chen, 2015***; ***Wrobel et al., 2015***). This phenomenon has been studied by the use of specific import mutants or the overexpression of artificial clogger proteins to induce a specific protein traffic jam at mitochondria. Indeed, the chronic impairment of mitochondrial protein import leads to cell death (***Boos et al., 2019***; ***Coyne et al., 2023***; ***Wang and Chen, 2015***; ***Wrobel et al., 2015***). Cells have developed a coordinated and multifaceted response network to counteract the accumulation of these precursor proteins. This response network is known as the mitoprotein-induced stress response and includes several pathways that were previously described independently (***Bock and Tait, 2020***; ***Boos et al., 2019***). Here, we show that this mitochondrial stress response is highly and specifically induced by a natural lipid compound, t-2-hex, providing evidence and a molecular mechanism of an endogenous modulation of mitochondrial protein import. As a consequence, the accumulated precursor proteins trigger the heat shock response, which increases the production of chaperones

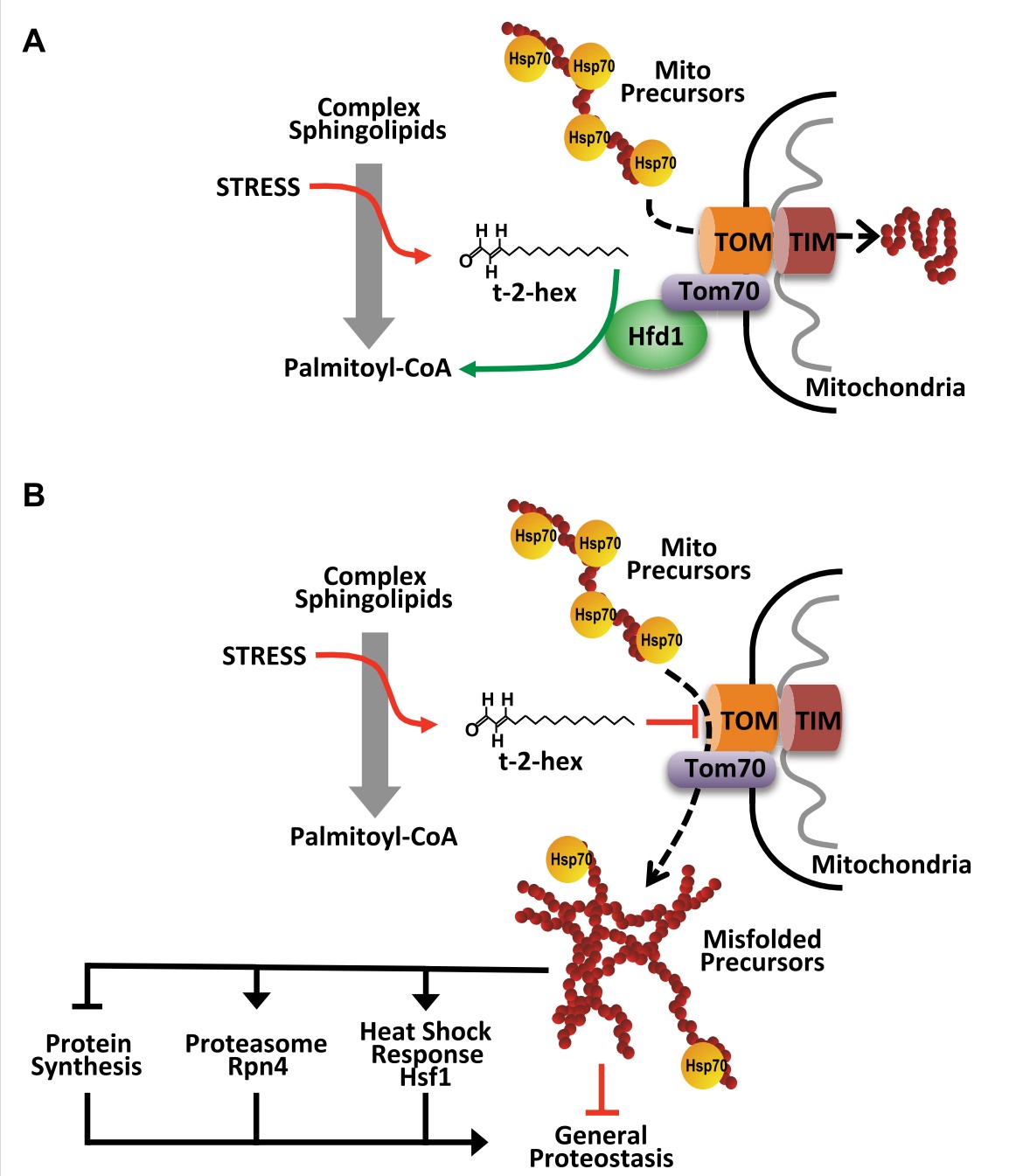

**Figure 9.** Model of the pro-apoptotic function of trans-2-hexadecenal (t-2-hex) and the anti-apoptotic function of Hfd1 at the mitochondrial Tom complex. Upper panel: Under normal conditions, the Hfd1 lipid aldehyde dehydrogenase located at the Tom complex safeguards mitochondrial protein import by t-2-hex degradation. Lower panel: Upon severe stress conditions or in the absence of Hfd1 function, an excess of t-2-hex directly lipidates Tom subunits such as Tom40 and thus inhibits mitochondrial pre-protein transport across the outer mitochondrial membrane. The resulting proteostatic imbalance in the cytosol, if not sufficiently repaired or counteracted by the heat shock response, proteasomal clearance or diminished de novo protein synthesis, can induce apoptotic cell death.

and enhances the proteolytic capacity of the ubiquitin-proteasome system (UPS) (**Boos et al., 2019**; **Weidberg and Amon, 2018**; **Wrobel et al., 2015**). This cellular response to mitochondrial dysfunction is also known as the unfolded protein response activated by mistargeting of proteins (UPRam). The heat shock transcription factor Hsf1 plays a central role in this response, as it is activated not only by heat stress but also during the transition from fermentation to respiration and in the presence of

mitochondrial import impairment, all of which result in the accumulation of mitochondrial precursors in the cytosol (**Boos et al., 2019**; **Groh et al., 2023**; **Hahn and Thiele, 2004**). The sequestration of cytosolic chaperones by these precursor proteins leads to the release and activation of Hsf1. This activation triggers a cascade of gene expressions, including the transcription factor Rpn4, which induces the expression of proteasomal genes to enhance UPS capacity. According to our results, lipidation of the TOM complex at various subunits upon pro-apoptotic conditions blocks the mitochondrial protein entry gate and triggers the heat shock response and proteasomal activity to counteract mislocalized protein accumulation in the cytoplasm. Several TOM subunits are lipidated by low t-2-hex doses in vitro, lack of function mutants in specific TOM subunits are resistant to t-2-hex, and t-2-hex inhibits specifically mitochondrial GFP activity and not its cytosolic counterpart independently of $\Delta\Psi$m changes, which indicates that the pro-apoptotic lipid directly inhibits mitochondrial protein import. It will be interesting to link specific t-2-hex lipidation events at the TOM complex with the apoptotic inhibition of mitochondrial protein import and cell death. Additionally, t-2-hex could directly act on mitochondrial pre-proteins or cytosolic chaperones rendering the precursors aggregation prone. However, this seems to be less likely as the mitochondrial and cytosolic GFP used in the import studies here only differ in the small and cysteine-free Pre Su9 pre-peptide.

The external addition of the pro-apoptotic lipid aldehyde t-2-hex is an efficient experimental model to study the induction of apoptosis in various cell types or in purified mitochondria (**Amaegberi et al., 2019**; **Chipuk et al., 2012**; **Kumar et al., 2011**; **Manzanares-Estreder et al., 2017**). The internal trigger of apoptosis comes from the sphingolipid degradation pathway, which is induced upon different cellular stress situations. Genetic manipulations to either favor production of t-2-hex or avoid its further degradation cause mitochondrial damage and inhibit proliferation (**Manzanares-Estreder et al., 2017**). The first enzymatic steps in this pathway from ceramide to sphingosine-1-phosphate and t-2-hex altogether take place in the ER. The t-2-hex detoxifyer Hfd1 localizes in foci along the mitochondrial network (**Manzanares-Estreder et al., 2017**) and associates with the outer membrane of the organelle probably by interacting with TOM. Here, Hfd1 would serve as a surveillance factor in order to avoid an impairment of pre-protein import caused by increasing lipidation of the TOM entry gate. This is in line with recent findings, which reveal association of the core TOM complex with an increasing variety of quality control factors such as chaperones, ubiquitin ligases, deubiquitilases, and proteases to constantly monitor the protein import process at the mitochondrial surface (**Opaliński et al., 2018**; **Schulte et al., 2023**).

Mitochondrial protein import is a tightly monitored traffic hub in eukaryotic cells. Its prolonged malfunction leads to cell death and subtle perturbations cause diseases including neurodegeneration (**Coyne et al., 2023**). Insufficient clearance of stalled mitochondrial precursor proteins from the TOM complex leads to their accumulation in the cytosol and more generally, impairments in mitochondrial protein import lead to the formation of aberrant cytosolic deposits containing high-risk precursor and immature mitochondrial proteins. These deposits further accumulate and cause aggregation of disease-related proteins, including α-synuclein and amyloid β, resulting in cytosolic protein homeostasis imbalance (**Nowicka et al., 2021**). Here, we show that this proteostatic imbalance is efficiently created by the t-2-hex lipid aldehyde, which cannot be counteracted by proteasomal clearance given its inefficiency in breaking down protein aggregates. This provides evidence that pro-apoptotic lipid signaling produces a collapse of protein homeostasis via mitochondrial protein import defects. Of note, the capacity of mitochondrial protein import has been recently directly correlated with the clearance of toxic protein aggregates in the cytosol (**Ruan et al., 2017**; **Schlagowski et al., 2021**), further strengthening the concept that mitochondrial protein import is a crucial modulator of general proteostasis and specific detoxification of protein aggregates (**Maruszczak et al., 2023**). In line with the results presented in this study, the levels of pro-apoptotic lipid effectors such as t-2-hex are decisive for the entry into apoptotic death by increasing aggregate formation. Apart from triggering a cell death decision, this balance might also be crucial for cellular lifespan. A response to a mild mitochondrial protein import defect extends lifespan in *Caenorhabditis elegans* (**Sladowska et al., 2021**), and more generally, proteasomal activity or a reduced or better controlled protein synthesis rate correlate with longevity in yeast cells (**Kruegel et al., 2011**; **Martinez-Miguel et al., 2021**; **Mittal et al., 2017**; **Steffen et al., 2008**). Given that a slowed protein synthesis rate is also beneficial to counteract the pro-apoptotic function of t-2-hex, the action of anti-apoptotic enzymes such as Hfd1 could have effects on cellular lifespan as well. It will be crucial for future studies to evaluate the impact

of mitochondrial protein import during lipid-mediated cell death in higher eukaryotic models, which has the potential to significantly broaden our comprehension of the mechanisms leading to intrinsic cell death and their regulation.

## Materials and methods

### Yeast strains and growth conditions

The yeast strains used in this work are detailed in *Supplementary file 7*. Yeast cultures were grown, unless otherwise indicated, at 28 °C in Yeast Extract Peptone Dextrose (YPD) media containing 2% glucose, Yeast Extract Peptone Galactose (YPGal) media containing 2% galactose, or in Synthetic Dextrose (SD) media containing 0.67% yeast nitrogen base with ammonium sulfate and without amino acids, 50 mM succinic acid (pH 5.5) and 2% of the respective sugar. According to specific auxotrophies of individual strains, methionine (10 mg/l), histidine (10 mg/l), leucine (10 mg/l), or uracil (25 mg/l) were supplemented. Yeast cells were transformed with the lithium acetate/PEG method described by *Gietz and Schiestl, 2007*. Strains expressing particular proteins as Tap-fusions from the chromosomal locus are described in *Ghaemmaghami et al., 2003*. Genomic deletion of particular genes was done with PCR-based methods using KAN-MX or his5[+] (*S. pombe*) containing plasmids pUG6 or pUG27 (*Gueldener et al., 2002*). All deletion strains were verified with diagnostic PCR on purified genomic DNA. For genomic promoter swapping, a KAN-MX::TDH3p containing construct was made by cloning the *TDH3* promoter in the pUG6 plasmid at the *Eco*RV and *Spe*I sites. The Hfd1 constitutive overexpressor strain was made by replacing the native promoter with the KAN-MX::TDH3p cassette.

Bioactive lipids t-2-hex and t-2-hex-H$_2$ were purchased from Sigma and applied directly to yeast cultures from 1 mg/ml stock solutions in DMSO. Cycloheximide (Sigma) was added to a final concentration of 250 µg/ml from a 50 mg/ml stock solution. The ribosomal inhibitor (2-(2-Bromophenyl))–2, 3-dihydro-1H-naphtho[1,8-de][1,3,2]diazaborine (DAB) was purchased from TCI Europe and applied to yeast cultures at a final concentration of 20 µg/ml from a 20 mg/ml stock in ethanol. Mitochondrial uncoupler CCCP was purchased from Sigma and applied directly to yeast cultures to a final concentration of 20 µM from stock solutions in DMSO.

### Plasmids

The single-copy destabilized reporter plasmid pAG413-GRE2-lucCP[+] was described previously (*Rienzo et al., 2012*). The HFD1p-lucCP[+] reporter was constructed by cloning the entire *HFD1* promoter (−975/–4) into the *Sac*I and *Sma*I sites of pAG413-lucCP[+] (*Rienzo et al., 2012*). For the assay of specific cis-regulatory elements, synthetic oligonucleotides containing three repetitions of PACE and four repetitions of HSE sequences were inserted into the *Bsp*EI site of plasmid pAG413-CYC1Δ-lucCP[+] (*Rienzo et al., 2012*). The sequences used were the following (consensus sequence for each TF binding site is underlined): PACE, 5′-CCGG<u>CGGTGGCAAA</u>GATAT<u>CGGTGGCAAA</u>GTAAT<u>CGGT</u><u>GGCAAA</u>T-3′; HSE, 5′-CCGGCGATAT<u>CTTC</u>TA<u>GAA</u>GC<u>TTC</u>TA<u>GAA</u>GT-3′. Other TF-specific real-time luciferase reporter plasmids (AP-1, CRE, PDRE) used in this study were described previously (*Dolz-Edo et al., 2013*; *Vanacloig-Pedros et al., 2019*). A mitochondrially targeted GFP fusion encoded in the pVT100-mtGFP was used to visualize mitochondrial morphology in yeast cells (*Westermann and Neupert, 2000*). For flow cytometric determination of galactose inducible cytosolic or mitochondrial GFP, the centromeric plasmids pRS416-GAL1-GFP or pYX113-mtGFP were used (*Fita-Torró et al., 2023*; *Westermann and Neupert, 2000*).

### Quantitative growth assays

For the quantitative estimation of growth parameters, fresh yeast overnight precultures of the indicated yeast strains were diluted in triplicate in the assay medium in multiwell plates to a starting OD$_{600}$=0.1. Growth was then continuously monitored at a wavelength of 600 nm every 30 min at 28 °C on a Tecan Spark multiplate reader for the indicated times and with two rounds of orbital shaking in between reads. The growth curves were processed in Microsoft Excel, and doubling times, lag times, and growth yields were calculated with the PRECOG software (*Fernandez-Ricaud et al., 2016*) with respect to control conditions.

### Preparation of total protein extracts

Yeast whole cell extracts were generated using the boiling method of alkaline pretreated cells described by *Kushnirov, 2000*. A total of 2.5 OD$_{600}$ were processed for each individual extract.

## Western blot

The following primary antibodies were used for the immunological detection of specific proteins on PVDF membranes: anti-Tap (CAB1001 rabbit polyclonal antibody, Invitrogen, 1:10,000), anti-HA (12CA5 mouse monoclonal antibody, Roche, 1:10,000), anti-Ubiquitin (P4D1 mouse monoclonal antibody, Santa Cruz Biotechnology, 1:1000), anti-Pgk1 (22C5D8 mouse monoclonal antibody, Invitrogen, 1:5000). Immunoblots were ECL developed using the secondary antibodies anti-rabbit IgG HRP (Cytiva NA934, 1:10,000) and anti-mouse IgG HRP (Cytiva NA931, 1:10,000). Blots were visualized with an Image Quant LAS4000 system. Total protein loading was visualized with the Direct Blue 71 method described by *Hong et al., 2000*.

## Flow cytometric quantification of mt-GFP import and mitochondrial membrane potential (ΔΨm)

Yeast wild-type or *Δrpn4* cells harboring single-copy GAL1p-GFP were pre-grown in synthetic raffinose medium (SRaf) without uracil to exponential growth phase. Cells were then diluted to $OD_{600}$=0.1 and induced with 1% galactose for 2 hr in the presence or absence of the indicated t-2-hex or t-2-hex-$H_2$ concentrations. Cell aliquots were passed through a MACS Quant 10 flow cytometer (Miltenyi Biotec). GFP was excited with a 488 nm laser and emission was detected by applying a 525/550 nm band pass filter set. For ΔΨm determinations, yeast wild-type cells were pre-grown in synthetic galactose medium (SGal) to exponential phase and treated for 2 hr with the indicated CCCP or t-2-hex concentrations. Cells were then washed, resuspended in 0.25 ml PBS and supplemented with 2.5 µM rhodamine 123 for another 30 min. PBS-washed cells were passed through the flow cytometer with the above indicated filter selection. All GAL1p-GFP and rhodamine 123 experiments were performed on three independent biological samples. 20,000 cells were analyzed for each measurement, and aggregated cells (<5%) were excluded from further analysis.

## Co-immunoprecipitation experiments

Yeast cells expressing full-length Hfd1 fused at the C-term with HA from expression vector pAG426-GPD-ccdB-HA and harboring endogenous genomic fusions of *TOM22* or *TOM70* with the TAP epitope were grown in selective SD medium to $OD_{600}$=1. Cells were washed 1 x in cold $H_2O$ and 1 x in cold buffer A (50 mM Tris/HCl pH 7.5, 15 mM EDTA, 150 mM NaCl, 2 mM DTT, 0.1% Triton X-100, 1 mM PMSF). Cells were lysed by mechanical disruption with 0.5 mm glass beads in buffer A supplemented with complete protease inhibitor cocktail (Roche). Equal amounts of total protein (input samples), quantified by the Bradford method, were incubated with Pan Mouse IgG Dynabeads (Invitrogen) for 2 hr at 4 °C. Beads were extensively washed (5 x) with buffer A and finally resuspended in 2 x Laemmli buffer. Input and co-precipitated samples were analyzed by anti-HA and anti-TAP immunological detection.

## Real-time luciferase assays

Time-elapsed assays with destabilized firefly luciferase reporters in living yeast cells were performed essentially as described before (*Rienzo et al., 2012*). Yeast strains containing the indicated luciferase fusion genes on centromeric plasmids were grown at 28 °C in Synthetic Dextrose (SD) medium lacking histidine at pH = 3 to low exponential phase. Culture aliquots were then incubated with 0.5 mM luciferin (free acid, Synchem, Germany) on a roller at 28 °C for 1 hr. The cells were then transferred in 135 µl aliquots to white 96 well plates (Nunc) containing the indicated concentrations of stressor or solvent. The light emission was continuously recorded on a GloMax microplate luminometer (Promega) in three biological replicas. Data were processed with Microsoft Excel. Raw data were normalized for the number of cells in each individual assay and set to 1 for time point 0. For the comparison of induction folds, luciferase activities are shown throughout this work as relative light units (R.L.U). For the comparison of induction sensitivities, the maximal luciferase activity was set to 100% for the most responsive stressor concentration.

## Fluorescence microscopy

Exponentially growing yeast cells on SD media were visualized on a Leica SP8 confocal microscope with a HCXPL APO CS2 63 x objective or on a Zeiss LSM980 confocal microscope with a 63 x objective and Airyscan detector. GFP was visualized with 488 nm excitation and 509 nm emission wavelengths.

For cytoplasmic protein aggregation assays, yeast strains with endogenously GFP-tagged *HSP104* were used. For the visualization of global mitochondrial morphology, yeast strains with plasmid-encoded mtGFP (pVT100-mtGFP) targeted to the mitochondrial lumen were used.

## Quantification of proteasome activity

Chymotrypsin–like activity of the proteasome was measured as described previously (*Kruegel et al., 2011*) with some variations. 10–20 mL of yeast culture were grown to $OD_{600}$ 1–2 and treated with DMSO, t-2-hex, or t-2-hex-$H_2$ for 3 h. Before cell lysis, cells were washed 1 x with ice-cold water and 1 x with cold lysis buffer (50 mM Tris-HCl, pH 7.5, 0.5 mM EDTA, 5 mM $MgCl_2$). Cells were resuspended in 500 µL of ice-cold lysis buffer supplemented with cOmplete Mini, EDTA-free protease inhibitor cocktail (Roche), and transferred to 2 mL tubes with screw cap. After addition of glass beads (0.5 mm diameter), cells were lysed in the cold with a Precellys Evolution Touch homogenizer (Bertin Technologies) repeating five cycles of 30 s at 10,000 rpm and 30 s in ice. Cell extracts were cleared at 5000 g for 5 min at 4 °C and protein concentration was quantified by the Bradford assay (BioRad). Finally, proteasomal activity was assessed using the Suc-LLVY-AMC substrate (Bachem). 50 µg of total protein extract were diluted with lysis buffer in a final volume of 190 µl in 96-well black microtiter plates. After addition of 10 µL of 2 mM stock solution of Suc-LLVY-AMC substrate, fluorescence intensity was measured during 60 min at 30 °C with a Tecan Spark microplate reader (Tecan) with an excitation wavelength of 380 nm and an emission wavelength of 460 nm. Chymotrypsin–like activity was calculated as the slope of the linear part of the fluorescence intensity curve and expressed as the % of the Control condition.

## RNA seq experiments

30 mL of *hfd1Δ* exponentially growing cells were treated in triplicate with DMSO or 100 µM t-2-hex for 3 hr. After treatment, cells were collected, washed with ice-cold water, and frozen in liquid nitrogen. Total RNA extraction was performed using the acid phenol method. Cell pellets were resuspended in 400 µL of TES buffer (10 mM Tris/HCl pH 7.5, 10 mM EDTA, 0.5% SDS) and an equal volume of acid phenol mix (saturated with 0.1 M citrate buffer pH 4.3). The mixture was incubated at 65 °C for 45 min. After 5 min on ice, phases were separated by centrifugation (1 min, 10,000 rpm), and the aqueous phase was transferred to a new tube for another extraction with acid phenol. Again, the aqueous phase was transferred to another tube containing 40 µL of 3 M NaAcetate pH 5.0 and 2.5 vol of 100% ethanol. Finally, RNA was pelleted by centrifugation at 14,000 rpm for 10 min, washed once with 70% ethanol, and resuspended in 100 µL of Milli-Q water. Following extraction, 50 µg of total RNA were DNase I digested and purified using the RNeasy Mini Kit (QIAGEN) according to the manufacturer's instructions. RNA integrity was assessed using a 1% agarose gel.

The samples were processed and massively sequenced by BGI Genomics using the BGISEQ platform (https://www.bgi.com/us/sequencing-services/rna-sequencing-solutions/transcriptome-sequencing/). RNA integrity was evaluated using the Agilent Bioanalyzer 2100 (Agilent Technologies) and libraries constructed following DNBSEQ Eukaryotic Stranded Transcriptome library preparation protocol. Samples were sequenced at paired-end 100 bp read length in the DNBSEQ platform. The sequencing data were filtered using SOAPnuke software (version 1.5.2) (*Cock et al., 2010*) and clean reads were mapped with HISAT2 (version 2.0.4) (*Kim et al., 2015*) to the reference genome (*Saccharomyces_cerevisiae_S288C* version GCF_000146045.2_R64). On average, about 47.1 million clean reads were generated per sample, and the average mapping ratio with the yeast reference genome was 95.7%, with an average mapping ratio with particular genes of 75.0%. A total of 5851 genes was identified.

All RNAseq samples used in this study are available to download from ArrayExpress (https://www.ebi.ac.uk/biostudies/arrayexpress/studies) under dataset accession number E-MTAB-13188. For Gene Ontology (GO) analysis of transcriptomic data, genes upregulated (>2 $log_2$FC) and downregulated (<-2 $log_2$FC) by t-2-hex were selected. A total of 675 up-regulated genes and 530 down-regulated genes were subjected to GO category enrichment analysis using ShinyGo 0.80 software (http://bioinformatics.sdstate.edu/go/) (*Cohen et al., 2020*). The significantly enriched functional groups are summarized in *Supplementary file 1* (up-regulated genes) and *Supplementary file 2* (down-regulated genes).

### In silico analysis of enriched promoter motifs

RNAseq raw data were analyzed with the Integrated System for Motif Activity Response Analysis (ISMARA) software, powered by the SwissRegulon Portal from the Swiss Institute of Bioinformatics *Balwierz et al., 2014*. ISMARA is an online tool for automatically inferring regulatory networks from gene expression data and directly processes raw data of a set of samples and identifies the key transcription factors and miRNAs driving the observed expression state changes.

In this study, RNAseq FASTQ files of the different samples (3 x control, 3 x t-2-hex) were uploaded to ISMARA web portal (https://ismara.unibas.ch/mara/) and analyzed against sacSer2 genome assembly. After data processing, results were downloaded for further analysis.

### Saturated transposon analysis (SATAY)

SATAY was essentially performed as originally described in *Michel et al., 2017* with some modifications.

#### Preparation of transposon mutant library

The *By*K352 *hfd1Δ::KanMX* strain was transformed with pBK549 plasmid containing the transposon (*Michel and Kornmann, 2022*). Transformed cells were inoculated into 200 mL of SC liquid +0.2% Glucose +2% Raffinose -Uracil at $OD_{600}$ of 0.15 and the culture was grown until saturation ($OD_{600}$=3.9) for 16 hr. The culture was then concentrated by centrifugation to obtain a final $OD_{600}$ of 39. From this culture, 200 µL were plated on 50×8.5 cm plates containing 25 mL of SC +2% galactose-adenine using glass beads. Plates were incubated for 3 wk at 30 °C. After incubation, clones reached a density on the plate of 46750 colonies/plate, on average, with a total number of 2.3E+06 generated mutants. All colonies were then pooled together with very low volume of sterile water. From the pool, two independent cultures of 1 L were inoculated in SC +2% Glucose -Adenine. The control one was inoculated at an initial $OD_{600}$ of 0.2 and grown until saturation ($OD_{600}$ of 8) at 30 °C for 36 hr. The other culture, used for t-2-hex treatment, was grown for 36 hr from an initial $OD_{600}$ of 0.05–3, and then diluted to $OD_{600}$ 0.1 in 500 mL of SC +2% Glucose -Adenine treated with 50 µM of t-2-hex (Avanti Polar Lipids). After growing for 32 hr to OD600 1.1, it was diluted again to $OD_{600}$ 0.1 in 500 mL SC +2% Glucose -Adenine+50 µM of t-2-hex and grown for 24 hr to saturation (OD600 2.9). Finally, cells of both cultures (Control and t-2-hex treatment) were harvested for genomic DNA extraction.

#### Genomic DNA library preparation

A 500 mg cell pellet was resuspended in 500 µL of cell breaking buffer (2% Triton X-100, 1% SDS, 100 mM NaCl, 100 mM Tris-HCl pH 8.0, 1 mM EDTA) and divided into 280 µL aliquots. To each aliquot, 200 µL of Phenol:Chloroform:Isoamyl alcohol (25:24:1) and 300 µL of unwashed glass beads (0.5 mm) were added. Cells were lysed by subjecting them to 10 cycles of vortexing for 30 s followed by incubation on ice for 30 s. Next, 200 µL of TE buffer was added to each lysate, which was then centrifuged at 16,000 g for 5 minat 4 °C. The upper layer (approximately 400 µL) was carefully transferred to a new tube, and 2.5 vol of 100% ethanol were added, followed by mixing by inversion. The DNA was pelleted by centrifugation at 16,000 g for 5 min at 20 °C. The supernatant was discarded, and the DNA pellets were resuspended in 200 µL of RNAse A solution (250 µg/ml) and incubated for 15 min at 55 °C with gentle agitation using a thermoblock at 1000 rpm. Then, 2.5 vol of 100% ethanol and 0.1 vol of 3 M NaOAc pH 5.2 were added. The DNA was pelleted by centrifugation at 16,000 g for 5 min at 20 °C. The pellets were washed with 70% ethanol and dried for 10 min at 37 °C. Finally, the pellets were resuspended in a total volume of 100 µl $H_2O$ and incubated for 10 min at 55 °C with gentle agitation using a thermoblock at 700 rpm. The DNA was then analyzed on a 0.6% 1 x TBE agarose gel using a standard 1 kb GeneRuler, and quantification was performed using Fiji software.

#### Library amplification and sequencing

Next, 2 µg of both genomic DNA were digested using 50 U of *Mbo*I (Thermo Fisher Scientific) and *Hin*1II (Thermo Fisher Scientific) enzymes at 37 °C for 24 hr, in separated reactions. Subsequently, the enzymatic reactions were heat-inactivated at 65 °C for 20 min. The resulting DNA fragments were circularized by adding 25 Weiss U of T4 ligase (Thermo Fisher Scientific) and ligated at 22 °C for 6 hr in a total volume of 400 µL. To precipitate the circularized DNA molecules, a mixture of 0.3 M sodium acetate (pH 5.2), 1 mL of ethanol, and 5 µg of linear acrylamide (Ambion AM9520) was applied. The

mixture was then incubated at –20 °C overnight. After precipitation, the DNA was pelleted at 16,000 g and 4 °C for 20 min, followed by washing with 1 mL of 70% ethanol. The DNA was then dried at 37 °C for 10 min. The circularized DNA samples were resuspended in 100 µL of water. For transposon amplification, PCR was performed using the forward primer P5_MiniDs (AATGATACGGCGACCACCGA GATCTACtccgtcccgcaagttaaatatga) and the reverse primer P7_MiniDS (CAAGCAGAAGACGGCA TACGAGATNNNNNNNNacgaaaacgaacgggataaatac). Each primer consisted of two parts: the Illumina adaptor (uppercase letters) and a fragment from MiniDs (lowercase letters). The 8-nucleotide index present in P7_MiniDs allows distinction among different libraries. For PCR, circularized DNA was divided into 10×100 µL reactions containing: 10 µL of ligated DNA, 10 µL dNTPs mix 2 mM each, 10 µL 10 x DreamTaq Buffer (Thermo Fisher Scientific), 1 µL 100 uM primer P5, 1 uL 100 µM primer P7, 2 µL DreamTaq DNA Polymerase (5 U/µL) (Thermo Fisher Scientific). PCR was performed under the following conditions: 95 °C 5 min, 35 x [95 °C 30 s, 55 °C 30 s, 72 °C 3 min], 72 °C 10 min. Finally, for each condition (Control, t-2-hex) all PCR reactions were pooled into one *Hin*1II-digested pool and one *Mbo*I-digested pool. 100 µL from each pool were purified with CleanNGS magnetic beads (CleanNA) according to manufacturer instructions.

Primer Name Sequence (5′ → 3′):

P5_MiniDs AATGATACGGCGACCACCGAGATCTACtccgtcccgcaagttaaatatga
P7_MiniDs_i1 CAAGCAGAAGACGGCATACGAGATCGAGTAATacgaaaacgaacgggataaatac
P7_MiniDs_i2 CAAGCAGAAGACGGCATACGAGATTCTCCGGAacgaaaacgaacgggataaatac
P7_MiniDs_i3 CAAGCAGAAGACGGCATACGAGATAATGAGCGacgaaaacgaacgggataaatac
P7_MiniDs_i4 CAAGCAGAAGACGGCATACGAGATGGAATCTCacgaaaacgaacgggataaatac

Purified PCR products were then checked with 4200 TapeStation System (Agilent Technologies) and quantified by fluorimetry with Qubit dsDNA HS Assay Kit (Invitrogen) and Qubit 4 Fluorometer (Invitrogen). Equal amounts of *Mbo*I-digested and *Hin*1II-digested libraries from both Control and t-2-hex samples, were pooled, diluted to 20 pM, and sequenced at single 75 bp read length using MiSeq Reagent Kit v3 150-cycles (Illumina) on the MiSeq platform (Illumina), according to manufacturer instructions. 3.4 µL of 100 µM primer 688_minidsSEQ1210 (TTTACCGACCGTTACCGACCGTTT TCATCCCTA) and 600 µL of 0.5 µM primer Custom_index1 (GGTTTTCGATTACCGTATTTATCCCGTT CGTTTTCGT) were loaded into wells 12 and 19 of the sequencing cartridge, respectively.

*Sequences alignment and determination of transposon insertion sites and corresponding read numbers*: The FastQ files were analyzed, aligned to the sacCer3 genome, and processed using R and Bioconductor packages ShortRead, Rsamtools, and Rbowtie (**Michel and Kornmann, 2022**, **Supplementary files 1 and 2**). Volcano plots were generated from two technical replicates of the sequencing. All SATAY data from this study are deposited to download from ArrayExpress (https://www.ebi.ac.uk/biostudies/arrayexpress/studies) under dataset accession number E-MTAB-13208.

## t-2-hex-alkyne in vitro lipidation assays

Lipidation assays with the t-2-hex-alkyne probe were essentially performed as described previously (**Jarugumilli et al., 2018**). Yeast cells expressing HA-tagged Tom40 from the natural promoter on centromeric plasmids (**Becker et al., 2011**) or control cells were grown to exponential growth phase in Yeast Extract Peptone Glycerol/Ethanol (YPGE) medium containing 3% glycerol/2% ethanol. Cells were washed 1 x with cold H$_2$O and 1 x with cold TSB buffer (10 mM Tris/HCl pH 7.5, 0.6 M sorbitol). Cells were lysed by mechanical rupture with 0.5 mm glass beads in TSB buffer supplemented with 1 mM PMSF and complete protease inhibitor cocktail (Roche). The mitochondrial fraction was enriched from the lysates by two rounds of differential centrifugation: 5 min at 3500 g discarding the pellet and subsequent centrifugation of the supernatant at 16,000 g for 10 min. The final mitochondria-enriched pellet was resuspended in 250 µl of Hepes lysis buffer (50 mM Hepes pH 7.4, 0.5% NP-40) supplemented with complete protease inhibitor cocktail (Roche). Total protein was quantified with the Bradford method, and the mitochondrial fractions adjusted to 0.1 mg/ml in Hepes lysis buffer. 500 µl aliquots were used for in vitro lipidation experiments. The indicated concentrations of t-2-hex-alkyne ((E)–2-Hexadecenal Alkyne, 20714 Cayman Chemicals) or solvent alone were added from a 1 mg/ml stock in ethanol and the samples were rotated at room temperature for 1 hr. In the thiol competition experiments, 1 mM DTT was added to the samples before the addition of t-2-hex-alkyne. After removal of the input samples (100 µl), the rest of the samples was treated with the

click reaction mix (40 µl) of 100 µM TBTA (678937 Sigma), 1 mM TCEP (C4706 Sigma), 1 mM CuSO$_4$, 100 µM Azide-PEG3-biotin conjugate (762024 Sigma) for 1 hr at room temperature. Samples were finally precipitated by adding 3.4 ml chilled methanol overnight at –80 °C. Proteins were pelleted by centrifugation at 17,000 g for 15 min and dried in a Speed Vac centrifuge for 5 min. Proteins were first dissolved by addition of 50 µl PBST (8 mM Na$_2$HPO$_4$, 150 mM NaCl, 2 mM KH$_2$PO$_4$, 3 mM KCl, 0.05% Tween 20, pH 7.4) containing 1% SDS and then diluted in 950 µl PBST. Protein mixtures were incubated with 40 µl PBST washed Streptavidin agarose resin (50% slurry, 69203–3 Merck) and incubated for 2 hr at room temperature on a roller. Agarose beads were washed 3 x with PBST and proteins eluted with 80 µl of elution buffer (95% formamide, 10 mM EDTA) for 5 min at 95 °C. The supernatant was supplied with 1 x Laemmli buffer and proteins were analyzed by immunoblotting with anti-HA antibodies.

For the chemoproteomic analysis of t-2-hex targets, the mitochondria-enriched fraction was prepared from wild-type cells as described above. A total of 50 µg of mitochondrial fraction was treated with 100 µM or 10 µM t-2-hex-alkyne or solvent alone and the samples were rotated at room temperature for 1 hr. Samples were click chemistry treated and methanol precipitated as described above. Dried protein pellets were dissolved first in 500 µl in resuspension buffer containing 50 mM Tris/HCl pH7.4, 150 mM NaCl, 10 mM EDTA, 1% SDS. Samples were further diluted by adding 500 µl of resuspension buffer containing 50 mM Tris/HCl pH7.4, 150 mM NaCl, 10 mM EDTA, 1% Brij97. 40 µl of PBS washed Streptavidin agarose beads were added to the samples and incubation was performed on a rotator at room temperature for 1.5 h. Beads were washed 3 x with PBS +0.2% SDS and 1 x with 250 mM ammonium bicarbonate. After removing the supernatant, beads were incubated with 500 µl of 8 M urea, 50 µl of 500 mM TCEP, and 50 µl of 400 mM iodoacetamide and incubated on a rotator in the dark for 40 min. Finally, beads were washed 2 x with 250 mM ammonium bicarbonate. Further sample preparation and proteomic analysis was performed in the proteomics facility of SCSIE University of Valencia, a member of Proteored. Proteins were digested on the beads with trypsin overnight. Tandem mass spectrometry analysis (LC–MS/MS) was performed in a Tims TOF fleX mass spectrometer (Bruker). The PASER system (Bruker) was used to search the MS and MSMS data with the Sequest algorithm (ProLuCID) with the following parameters: SwissProt 23.03.10 database; Trypsin specificity; IAM cys-alkylation and taxonomy not restricted.

An excel document was generated with all the proteins identified with FDR ≤1%.

Proteins were ranked according to their number of spectra assigned to the protein in the t-2-hex-alkyne versus mock- treated samples. Differential enrichment analysis of the proteomic data was performed with the Amica software (*Didusch et al., 2022*). Proteins were ranked according to their log$_2$ fold induction comparing lipid- and mock-treated samples with a threshold of ≥1.5, and the adjusted p-value was calculated. A total of 217 (100 µM t-2-hex) and 459 (10 µM t-2-hex) differentially enriched proteins were identified.

## Statistical analyses

Data are presented as mean ± standard error mean. The number of biological replicas (n) is given for each experiment in the figure legends. Comparisons between different groups were performed by the Student's t-test. All analyses were performed with GraphPad Prism 8.0.1 software (GraphPad Software, Inc). A p-value <0.05 was considered statistically significant.

## Acknowledgements

We would like to thank Susana Rodríguez-Navarro, Sergi Puig, Thomas Becker, Nils Wiedemann, and Markus Tamas for the kind gift of yeast strains. We thank Manuela Torres-Puente and Iñaki Comas for their technical assistance and help with the massive sequencing of SATAY libraries. We thank Luz Valero for technical assistance and help with the proteomic analysis of t-2-hex treated cell extracts. We thank Rosa Viana for assistance and help with confocal microscopy image acquisition. This work was funded by grants from Ministerio de Ciencia e Innovación PID2019-104214RB-I00 and PID2022-136371OB-I00 to AP-A and MP and by Wellcome Trust grant number 214291/Z/18/Z to BK. JLG-H was supported by a pre-doctoral fellowship from Generalitat Valenciana (ACIF/2021/171). LL-G was supported by a pre-doctoral fellowship from Generalitat Valenciana (ACIF/2022/232).

## Additional information

### Competing interests

Benoit Kornmann: Senior editor, *eLife*. The other authors declare that no competing interests exist.

### Funding

| Funder | Grant reference number | Author |
|--------|------------------------|--------|
| Ministerio de Ciencia e Innovación | PID2019-104214RB-I00 | Amparo Pascual-Ahuir Markus Proft |
| Wellcome Trust | 214291/Z/18/Z | Benoit Kornmann |
| Generalitat Valenciana | ACIF/2021/171 | José Luis Garrido-Huarte |
| Generalitat Valenciana | ACIF/2022/232 | Lucía López-Gil |
| Ministerio de Ciencia e Innovación | PID2022-136371OB-I00 | Amparo Pascual-Ahuir Markus Proft |

The funders had no role in study design, data collection and interpretation, or the decision to submit the work for publication. For the purpose of Open Access, the authors have applied a CC BY public copyright license to any Author Accepted Manuscript version arising from this submission.

### Author contributions

Josep Fita-Torró, Data curation, Formal analysis, Investigation, Methodology, Writing - original draft, Writing - review and editing; José Luis Garrido-Huarte, Lucía López-Gil, Agnès H Michel, Data curation, Investigation, Writing - original draft, Writing - review and editing; Benoit Kornmann, Data curation, Funding acquisition, Writing - original draft, Writing - review and editing; Amparo Pascual-Ahuir, Conceptualization, Supervision, Funding acquisition, Investigation, Writing - original draft, Writing - review and editing; Markus Proft, Conceptualization, Data curation, Formal analysis, Supervision, Funding acquisition, Investigation, Methodology, Writing - original draft, Writing - review and editing

### Author ORCIDs

Benoit Kornmann (ID) https://orcid.org/0000-0002-6030-8555
Markus Proft (ID) https://orcid.org/0000-0002-6788-5830

Reviewer #3 (Public review): https://doi.org/10.7554/eLife.93621.4.sa1
Author response https://doi.org/10.7554/eLife.93621.4.sa2

## Additional files

### Supplementary files

Supplementary file 1. Transcriptomic analysis (RNA seq) of t-2-hex-induced gene functions.

Supplementary file 2. Transcriptomic analysis (RNA seq) of t-2-hex repressed gene functions.

Supplementary file 3. Comparison of the transcriptomic response to mitochondrial import stress and t-2-hex overload.

Supplementary file 4. SATAY underenriched gene functions upon t-2-hex stress.

Supplementary file 5. SATAY enriched gene functions upon t-2-hex stress.

Supplementary file 6. Results of the chemoproteomic protein identification comparing t-2-hex-alkyne and solvent-treated cell extracts at 10 µM and 100 µM.

Supplementary file 7. Yeast strains used in this study.

MDAR checklist

### Data availability

All RNAseq data of this study are available to download from ArrayExpress (https://www.ebi.ac.uk/biostudies/arrayexpress/studies) under dataset accession number E-MTAB-13188. All SATAY data from this study are deposited to download from ArrayExpress (https://www.ebi.ac.uk/biostudies/

arrayexpress/studies) under dataset accession number E-MTAB-13208. All data generated or analysed during this study are included in the manuscript and supplemental tables; source data files are provided for Figures 4, 5 and 8 containing the original files for western blot analyses displayed in the figures.

The following datasets were generated:

| Author(s) | Year | Dataset title | Dataset URL | Database and Identifier |
|-----------|------|---------------|-------------|-------------------------|
| Fita-Torró J | 2023 | Uncovering novel genetic factors controlling sphingolipid-induced mitochondrial cell death in yeast through genome-wide transposon-insertion mutant screening (SATAY) | https://www.ebi.ac.uk/biostudies/ArrayExpress/studies/E-MTAB-13208 | ArrayExpress, E-MTAB-13208 |
| Fita Torró J | 2023 | Transcriptional response of *Saccharomyces cerevisiae* to the proapoptotic unsaturated fatty aldehyde t-2-hexadecenal by RNAseq | https://www.ebi.ac.uk/biostudies/ArrayExpress/studies/E-MTAB-13188 | ArrayExpress, E-MTAB-13188 |

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

# Appendix 1

## Appendix 1—key resources table

| Reagent type (species) or resource | Designation | Source or reference | Identifiers | Additional information |
|---|---|---|---|---|
| Strain, strain background (*Saccharomyces cerevisiae*) | BY4741 | EUROSCARF | | Genotype: *MATa; his3Δ1; leu2Δ0; met15Δ0; ura3Δ0* |
| Strain, strain background (*Saccharomyces cerevisiae*) | *hfd1Δ* | EUROSCARF | | Genotype: BY4741 with *hfd1Δ::KanMX* |
| Strain, strain background (*Saccharomyces cerevisiae*) | TDH3p-HFD1 | This study | | Genotype: BY4741 with *KanMX::pTDH3-HFD1* |
| Strain, strain background (*Saccharomyces cerevisiae*) | *rpn4Δ* | Sergi Puig collection | | Genotype: BY4741 with *rpn4Δ::KanMX* |
| Strain, strain background (*Saccharomyces cerevisiae*) | WT Hsp104-GFP | *Jacobson et al., 2012* | | Genotype: BY4741 with *HSP104-GFP::HIS3* |
| Strain, strain background (*Saccharomyces cerevisiae*) | *rpn4Δ* Hsp104-GFP | *Jacobson et al., 2012* | | Genotype: *rpn4Δ* with *HSP104-GFP::HIS3* |
| Strain, strain background (*Saccharomyces cerevisiae*) | Rpn8-TAP | *Ghaemmaghami et al., 2003* | | Genotype: BY4741 with *RPN8-TAP::HIS3* |
| Strain, strain background (*Saccharomyces cerevisiae*) | Aim17-TAP | *Ghaemmaghami et al., 2003* | | Genotype: BY4741 with *AIM17-TAP::HIS3* |
| Strain, strain background (*Saccharomyces cerevisiae*) | Aim17-TAP *hfd1Δ* | This study | | Genotype: Aim17-TAP with *hfd1Δ::KanMX* |
| Strain, strain background (*Saccharomyces cerevisiae*) | Aim17-TAP TDH3p-HFD1 | This study | | Genotype: Aim17-TAP with *KanMX::pTDH3-HFD1* |
| Strain, strain background (*Saccharomyces cerevisiae*) | Cox5a-TAP | *Ghaemmaghami et al., 2003* | | Genotype: BY4741 with *COX5A-TAP::HIS3* |
| Strain, strain background (*Saccharomyces cerevisiae*) | Ilv6-TAP | *Ghaemmaghami et al., 2003* | | Genotype: BY4741 with *ILV6-TAP::HIS3* |
| Strain, strain background (*Saccharomyces cerevisiae*) | Sdh4-TAP | *Ghaemmaghami et al., 2003* | | Genotype: BY4741 with *SDH4-TAP::HIS3* |

*Appendix 1 Continued on next page*

*Appendix 1 Continued*

| Reagent type (species) or resource | Designation | Source or reference | Identifiers | Additional information |
|---|---|---|---|---|
| Strain, strain background (*Saccharomyces cerevisiae*) | Mpc3-TAP | **Ghaemmaghami et al., 2003** | | Genotype: BY4741 with *MPC3-TAP::HIS3* |
| Strain, strain background (*Saccharomyces cerevisiae*) | Cyc7-TAP | **Ghaemmaghami et al., 2003** | | Genotype: BY4741 with *CYC7-TAP::HIS3* |
| Strain, strain background (*Saccharomyces cerevisiae*) | Cis1-TAP | **Ghaemmaghami et al., 2003** | | Genotype: BY4741 with *CIS1-TAP::HIS3* |
| Strain, strain background (*Saccharomyces cerevisiae*) | Tma10-TAP | **Ghaemmaghami et al., 2003** | | Genotype: BY4741 with *TMA10-TAP::HIS3* |
| Strain, strain background (*Saccharomyces cerevisiae*) | ByK352 | **Michel et al., 2017** | | Genotype: BY4741 with *ade2Δ::HIS3* |
| Strain, strain background (*Saccharomyces cerevisiae*) | ByK352 *hfd1Δ* | This study | | Genotype: ByK352 with *hfd1Δ::KanMX* |
| Strain, strain background (*Saccharomyces cerevisiae*) | *pdr1Δ* | Susana Rodríguez collection | | Genotype: BY4741 with *pdr1Δ::KanMX* |
| Strain, strain background (*Saccharomyces cerevisiae*) | *fmp52Δ* | Susana Rodríguez collection | | Genotype: BY4741 with *fmp52Δ::KanMX* |
| Strain, strain background (*Saccharomyces cerevisiae*) | *rps30bΔ* | Susana Rodríguez collection | | Genotype: BY4741 with *rps30bΔ::KanMX* |
| Strain, strain background (*Saccharomyces cerevisiae*) | *rpl16aΔ* | Susana Rodríguez collection | | Genotype: BY4741 with *rpl16aΔ::KanMX* |
| Strain, strain background (*Saccharomyces cerevisiae*) | *rps28aΔ* | Susana Rodríguez collection | | Genotype: BY4741 with *rps28aΔ::KanMX* |
| Strain, strain background (*Saccharomyces cerevisiae*) | *rpl40aΔ* | Susana Rodríguez collection | | Genotype: BY4741 with *rpl40aΔ::KanMX* |
| Strain, strain background (*Saccharomyces cerevisiae*) | Tom22-TAP | This study | | Genotype: BY4741 with *TOM22-TAP::HIS3* (**Ghaemmaghami et al., 2003**) and *hfd1Δ::KanMX* |

*Appendix 1 Continued on next page*

*Appendix 1 Continued*

| Reagent type (species) or resource | Designation | Source or reference | Identifiers | Additional information |
|---|---|---|---|---|
| Strain, strain background (*Saccharomyces cerevisiae*) | Tom70-TAP | This study | | Genotype: BY4741 with *TOM70-TAP::HIS3* (**Ghaemmaghami et al., 2003**) and *hfd1Δ::KanMX* |
| Strain, strain background (*Saccharomyces cerevisiae*) | Tom40-HA | **Becker et al., 2011** | | Genotype: YPH499 with *tom40Δ::ADE2* and plasmid pFL39-TOM40-HA |
| Strain, strain background (*Saccharomyces cerevisiae*) | Tom40 WT | **Qiu et al., 2013** | | Genotype: YPH499 with *tom40Δ::ADE2* and plasmid pFL39-TOM40-TRP1 |
| Strain, strain background (*Saccharomyces cerevisiae*) | Tom40 CFREE | **Qiu et al., 2013** | | Genotype: YPH499 with *tom40Δ::ADE2* and plasmid pFL39-TOM40CFREE-TRP1 |
| Strain, strain background (*Saccharomyces cerevisiae*) | *tom20Δ* | This study | | Genotype: BY4741 with *tom20Δ::KanMX* |
| Strain, strain background (*Saccharomyces cerevisiae*) | *tom70Δ* | This study | | Genotype: BY4741 with *tom70Δ::KanMX* |
| Strain, strain background (*Saccharomyces cerevisiae*) | Aim17-TAP *tom20Δ* | This study | | Genotype: Aim17-TAP with *tom20Δ::KanMX* |
| Strain, strain background (*Saccharomyces cerevisiae*) | HSP104-GFP | Markus Tamas | | Genotype: BY4741 with *HSP104-GFP::HIS3* |
| Strain, strain background (*Saccharomyces cerevisiae*) | HSP104-GFP *rpn4Δ* | Markus Tamas | | Genotype: BY4741 with *HSP104-GFP::HIS3 rpn4::KanMX* |
| Recombinant DNA reagent (plasmid) | pAG413-CYC1Δ-lucCP⁺ | **Rienzo et al., 2012** | | *AmpR*, *CEN*, luciferase, *HIS3* Destination vector for promoter cloning |
| Recombinant DNA reagent (plasmid) | pUG6 | **Güldener et al., 1996** | | AmpR Generation of loxP–KanMX–loxP gene disruption cassettes |
| Recombinant DNA reagent (plasmid) | pUG6-p*TDH3* | This study | | *AmpR* Generation of targeted *KanMx::*p*TDH3* insertion cassettes |
| Recombinant DNA reagent (plasmid) | pAG413-HFD1-lucCP⁺ | This study | | pAG413-lucCP⁺ with *HFD1* promoter |
| Recombinant DNA reagent (plasmid) | pAG413-GRE2-lucCP⁺ | **Rienzo et al., 2012** | | pAG413-lucCP⁺ with *GRE2* promoter |
| Recombinant DNA reagent (plasmid) | pAG413-4xHSE-lucCP⁺ | This study | | pAG413-lucCP⁺ with 4 repetitions of HSE element |
| Recombinant DNA reagent (plasmid) | pAG413-3xPACE-lucCP | This study | | pAG413-lucCP⁺ with 3 repetitions of PACE element |

*Appendix 1 Continued on next page*

*Appendix 1 Continued*

| Reagent type (species) or resource | Designation | Source or reference | Identifiers | Additional information |
|---|---|---|---|---|
| Recombinant DNA reagent (plasmid) | pAG413-3xPDRE-lucCP+ | *Vanacloig-Pedros et al., 2019* | | pAG413-lucCP+ with 3 repetitions of PDRE element |
| Recombinant DNA reagent (plasmid) | pAG413-3xAP-1-lucCP+ | *Dolz-Edo et al., 2013* | | pAG413-lucCP+ with 3 repetitions of AP-1 element |
| Recombinant DNA reagent (plasmid) | pAG413-3xCRE-lucCP+ | *Dolz-Edo et al., 2013* | | pAG413-lucCP+ with 3 repetitions of CRE element |
| Recombinant DNA reagent (plasmid) | pBK549 | *Michel and Kornmann, 2022* | | *AmpR*, *CEN*, Ac transposase under *GAL1* promoter, *ADE2* gene interrupted by the MiniDs transposon, *URA3* |
| Recombinant DNA reagent (plasmid) | pVT100-mtGFP | *Westermann and Neupert, 2000* | | *AmpR*, 2 µ, *ADH1* promoter, mitochondria-targeted GFP, *URA3* |
| Recombinant DNA reagent (plasmid) | pAG426-GPD-HFD1-HA | Markus Poft collection | | *AmpR*, 2 µ, *GPD* promoter, *HFD1* CDS with C-term HA tag, *URA3* |
| Recombinant DNA reagent (plasmid) | pFL39-TOM40-HA | *Becker et al., 2011* | | *AmpR*, *CEN*, *TOM40* promoter, *TOM40* CDS with C-term HA tag, *TRP1* |
| Recombinant DNA reagent (plasmid) | pFL39-TOM40WT | *Qiu et al., 2013* | | *AmpR*, *CEN*, *TOM40* promoter, Tom40 WT, *TRP1* |
| Recombinant DNA reagent (plasmid) | pFL39-TOM40CFREE | *Qiu et al., 2013* | | *AmpR*, *CEN*, *TOM40* promoter, Tom40 with mutated cysteines, *TRP1* |
| Recombinant DNA reagent (plasmid) | pRS416-GFP | *Fita-Torró et al., 2023* | | *AmpR*, *CEN*, *GAL1* promoter, GFP, *URA3* |
| Recombinant DNA reagent (plasmid) | pYX113-mtGFP | *Westermann and Neupert, 2000* | | *AmpR*, *CEN*, *GAL1* promoter, Pre-Su9-GFP, *URA3* |
| Sequence-based reagent (cloning primer) | BspEI-3PACE-1 | This study | Cloning PACE sites into pAG413-CYC1Δ-lucCP+ | Sequence 5' ->3': P-CCGGCGGTGGCAAAGATATCGG TGGCAAAGTAATCGGTGGCAAAT |
| Sequence-based reagent (cloning primer) | BspEI-3PACE-2 | This study | | Sequence 5' ->3': P-CCGGATTTGCCACCGATTACTTT GCCACCGATATCTTTGCCACCG |
| Sequence-based reagent (cloning primer) | BspEI-4HSE-1 | This study | | Sequence 5' ->3': P-CCGGCGATATCTTCTAGAAGCTTCTAGAAGT |
| Sequence-based reagent (cloning primer) | BspEI-4HSE-2 | This study | Cloning HSE sites into pAG413-CYC1Δ-lucCP+ | Sequence 5' ->3': P-CCGGACTTCTAGAAGCTTCTAGAAGATATCG |
| Sequence-based reagent (PCR primer) | HFD1-995SacI | This study | Cloning *HFD1* promoter from BY4741 into pAG413-CYC1Δ-lucCP+ | Sequence 5' ->3': GCCGAGCTCTGATGACAGTAATAACCAACTCG |
| Sequence-based reagent (PCR primer) | HFD1-1SmaI | This study | | Sequence 5' ->3': TCCCCCGGGGGTTGGTGATAAATTACTATGGCTATGGTTT |
| Sequence-based reagent (PCR primer) | TDH3pEcoRVFw | This study | | Sequence 5' ->3': GGCCGATATCATTATCAATACTGCCATTTC |
| Sequence-based reagent (PCR primer) | TDH3pSpeIRev | This study | Cloning *TDH3* promoter from BY4741 into pUG6 | Sequence 5' ->3': GGCCACTAGTTTGTTTGTTTATGTGTGTTTATTCG |

*Appendix 1 Continued on next page*

*Appendix 1 Continued*

| Reagent type (species) or resource | Designation | Source or reference | Identifiers | Additional information |
|---|---|---|---|---|
| Sequence-based reagent (PCR primer) | HFD1-Del-1 | This study | Amplification of *KanMX* disruption cassette from pUG6 for *hfd1Δ* strain generation | Sequence 5′ ->3′: AAAAGGAATATTCTAAAACCATAGCCATAGTAA TTTATCACCAACCAGCTGAAGCTTCGTACGC |
| Sequence-based reagent (PCR primer) | HFD1-Del-2 | This study | | Sequence 5′ ->3′: AGGTTACTTATACATCAAATAATTAATTAACCT TAAACATTACGTGCATAGGCCACTAGTGGATCTG |
| Sequence-based reagent (PCR primer) | HFD1-TDH3-1 | This study | Amplification of *KanMX*::p*TDH3* cassette from pUG6-pTDH3 for *HFD1* promoter replacement | Sequence 5′ ->3′: CAACGAATTTCCAGCCAAAAATTCCGAGTAGT TCATGATGAAAGAGCTGAAGCTTCGTACGCTGC |
| Sequence-based reagent (PCR primer) | HFD1-TDH3-2 | This study | | Sequence 5′ ->3′: AGACACTGGGGTATAATTCAATATTTTTGAGCC GTCGTTTGACATTTTGTTTGTTTATGTGTGTTTATTCG |
| Sequence-based reagent (PCR primer) | TOM20-Del-1 | This study | Amplification of *KanMX* disruption cassette from pUG6 for *tom20Δ* strain generation | Sequence 5′ ->3′: GAAACATTGCCTCAAGTGCCACCTTCATAAA GTTTATTTTCTATTCAGCTGAAGCTTCGTACGC |
| Sequence-based reagent (PCR primer) | TOM20-Del-2 | This study | | Sequence 5′ ->3′: GTAAAAGAAACAAAAACGGAGAAAAAAAGCA AGCAAAATGTTACTCGCATAGGCCACTAGTGGATCTG |
| Sequence-based reagent (PCR primer) | TOM70-Del-1 | This study | Amplification of *KanMX* disruption cassette from pUG6 for *tom70Δ* strain generation | Sequence 5′ ->3′: GATTCGGAAGTGAAATTACAGCTCACATCTAG GTTCTCAATTGCCACAGCTGAAGCTTCGTACGC |
| Sequence-based reagent (PCR primer) | TOM70-Del-2 | This study | | Sequence 5′ ->3′: TTAGTTTTTGTCTTCTCCTAAAAGTTTTTAAGTT TATGTTTACTGTGCATAGGCCACTAGTGGATCTG |
| Sequence-based reagent (PCR primer) | P5_MiniDs | This study | Forward primer for all libraries | Sequence 5′ ->3′: AATGATACGGCGACCACCGAGATCTAC tccgtcccgcaagttaaatatga |
| Sequence-based reagent (PCR primer) | P7_MiniDs_i1 | This study | Reverse primer for Control_Hin1II library | Sequence 5′ ->3′: CAAGCAGAAGACGGCATACGAGATCGA GTAATcgaaaacgaacgggataaatac |
| Sequence-based reagent (PCR primer) | P7_MiniDs_i2 | This study | Reverse primer for Control_MboI library | Sequence 5′ ->3′: CAAGCAGAAGACGGCATACGAGATTCTCC GGAacgaaaacgaacgggataaatac |
| Sequence-based reagent (PCR primer) | P7_MiniDs_i3 | This study | Reverse primer for t-2-hex_Hin1II library | Sequence 5′ ->3′: CAAGCAGAAGACGGCATACGAGATA ATGAGCGacgaaaacgaacgggataaatac |
| Sequence-based reagent (PCR primer) | P7_MiniDs_i4 | This study | Reverse primer for t-2-hex_MboI library | Sequence 5′ ->3′: CAAGCAGAAGACGGCATACGAGATGGA ATCTCacgaaaacgaacgggataaatac |
| Sequence-based reagent (PCR primer) | 688_minids SEQ1210 | This study | Sequencing primer | Sequence 5′ ->3′: TTTACCGACCGTTACCGACCGTTTTCATCCCTA |
| Sequence-based reagent (PCR primer) | Custom_ index1 | This study | Index identification | Sequence 5′ ->3′: GGTTTTCGATTACCGTATTTATCCCGTTCGTTTTCGT |
| Antibody | α-TAP (TAP Tag Rabbit Polyclonal) | Invitrogen | Product # CAB1001 | WB (1:5000) |
| Antibody | α-HA (HA Tag Mouse Monoclonal) | Proteintech | Cat No. 66006–2-Ig | WB (1:10000) |

*Appendix 1 Continued on next page*

*Appendix 1 Continued*

| Reagent type (species) or resource | Designation | Source or reference | Identifiers | Additional information |
|---|---|---|---|---|
| Antibody | α-Ubi (Ubiquitin P4D1 Mouse Monoclonal) | Santa Cruz Biotechnology | Cat No. SC-8017 | WB (1:1000) |
| Antibody | α-Pgk1 (PGK1 Mouse Monoclonal (22C5D8)) | Invitrogen | Product # 10073994 | WB (1:5000) |
| Antibody | α-rabbit (anti-rabbit IgG HRP) | Cytiva | Cat No. NA934 | WB (1:10000) |
| Antibody | α-mouse (anti-mouse IgG HRP) | Cytiva | Cat No. NA931 | WB (1:10000) |
| Chemical compound, drug | trans-2-hexadecenal (t-2-hex) | Avanti Research | 857459 P | |
| Chemical compound, drug | Hexadecanal (t-2-hex-H$_2$) | Avanti Research | 857458 M | |
| Chemical compound, drug | t-2-hex-Alkyne (E)–2-Hexadecenal Alkyne | Cayman Chemical | Item No. 20714 | |
| Chemical compound, drug | Diazaborine (DAB) (2-(5-Bromo-2-thienyl)–2,3-dihydro-1H-naphtho[1,8-de][1,3,2]diazaborine) | TCI Chemicals | Cat No. B3868 | 20 µg/mL |
| Chemical compound, drug | D-Luciferin free acid, 99% | Synchem | Product No. S039 | 0.5 mM |
| Chemical compound, drug | Suc-LLVY-AMC | Bachem | Product No. 4011369 | 0.1 mM |

