## [Editor Report · eLife Assessment]

This study is a **valuable** observation that deals with the toxic effects of an intermediary in lipid degradation [trans-2-hexadecenal (t-2-hex)] in yeast through modification of mitochondrial protein import via the TOM complex. We find that the claim that the TOM complex is a main target of t-2-hex are supported by **solid** evidence, however the molecular mechanism remains unclear allowing multiple interpretation. Despite the shortcomings, this study is inspiring for researchers from the organellar, protein trafficking and lipid field and serves as a starting point to further precise and mechanistic analyses of the phenomenon.

---

## [Referee Report · Reviewer #3 (Public review)]

Summary:

The authors investigate the effect of high concentrations of the lipid aldehyde trans-2-hexadecenal (t-2-hex) in a yeast deletion strain lacking the detoxification enzyme. Transcriptomic analyses as global read out reveals that a large range of cellular functions across all compartments are affected (transcriptomic changes affect 1/3 of all genes). The authors provide additional analyses, which indicate that mitochondrial protein import is affected.

Strengths:

Global analyses (transcriptomic and functional genomics approach) to obtain an overview of changes upon yeast treatment with high doses of t-2-hex.

Weaknesses:

The use of high concentrations of t-2-hex in combination with a deletion of the detoxifying enzyme Hfd1 limits the possibility to identify physiological relevant changes. For the follow-up analysis, the authors focus on mitochondrial proteins and describe an impairment of mitochondrial protein biogenesis, but the underlying molecular modification resulting in the observed impairment is not yet known.

---

## [Author Response]

The following is the authors’ response to the previous reviews

**Public Reviews:**

**Reviewer #2 (Public Review):**
This study elucidates the toxic effects of the lipid aldehyde trans-2-hexadecenal (t-2-hex). The authors show convincingly that t-2-hex induces a strong transcriptional response, leads to proteotoxic stress and causes the accumulation of mitochondrial precursor proteins in the cytosol.The data shown are of high quality and well-controlled. The genetic screen for mutants that are hyper-and hypo-sensitive to t-2-hex is elegant and interesting, even if the mechanistic insights from the screen are rather limited. Moreover, the authors show evidence that t-2-hex affects subunits of the TOM complex. However, they do not formally demonstrate that the lipidation of a TOM subunit is responsible for the toxic effect of t-2-hex. A t-2-hex-resistant TOM mutant was not identified. Nevertheless, this is an interesting and inspiring study of high quality. The connection of proteostasis, mitochondrial biogenesis and sphingolipid metabolism is exciting and will certainly lead to many follow-up studies.
**Reviewer #3 (Public Review):**
Summary:The authors investigate the effect of high concentrations of the lipid aldehyde trans-2-hexadecenal (t-2-hex) in a yeast deletion strain lacking the detoxification enzyme. Transcriptomic analyses as global read out reveal that a large range of cellular functions across all compartments are affected (transcriptomic changes affect 1/3 of all genes). The authors provide additional analyses, from which they built a model that mitochondrial protein import caused by modification of Tom40 is blocked.

Our initial transcriptomic study with high doses of t-2-hex in a detoxifying mutant as an experimental approach is only a starting experiment and was aimed to identify as many determinants of t-2-hex toxicity as possible as stated in the manuscript. From this, we developed multiple independent approaches in wild-type (and mutant) cells at low t-2-hex concentrations, demonstrating that proteostasis and mitochondrial protein trafficking are physiologically important targets of the pro-apoptotic lipid. Specifically, proteostasis-specific PACE reporters are robustly induced in a detoxification mutant by 5mM t-2-hex (Figure 3D,E) and significantly induced by 10 mM t-2-hex in detoxification competent wild type cells (new Figure 3F).

We do not propose Tom40 as the lipid's primary target, while we show that several subunits of the TOM (and TIM) complex are directly targeted by low t-2-hex concentrations in vitro (Figure 8B), and Tom20 and Tom70 are important for lipid toxicity (Figure 8D) and mitochondrial protein trafficking in vivo (Suppl. Figure 2).

Strengths:Global analyses (transcriptomic and functional genomics approach) to obtain an overview of changes upon yeast treatment with high doses of t-2-hex.Weaknesses:The use of high concentrations of t-2-hex in combination with a deletion of the detoxifying enzyme Hfd1 limits the possibility to identify physiological relevant changes. From the hundreds of identified targets the authors focus on mitochondrial proteins, which are not clearly comprehensible from the data.

The initial transcriptomic study with high doses of t-2-hex in a detoxifying mutant is a starting experiment and was aimed to identify as many determinants of t-2-hex toxicity as possible as stated in the manuscript. As stated (page 4), genes up-regulated (>2 log2FC) by t-2-hex were selected and subjected to GO category enrichment analysis (Supplemental Table 1). We found that “Mitochondrial organization” was the most numerous GO group activated by t-2-hex. Among the strongly t-2-hex induced genes encoding mitochondrial proteins, CIS1 represented the most inducible gene with a known mitochondrial function. Cis1 is the central protein of the MitoCPR pathway, which is specifically induced upon and protects from mitochondrial protein import stress. We further show that proteostasis and mitochondrial protein trafficking are physiologically important targets at low t-2-hex doses in several independent experimental approaches: proteostasis-specific PACE reporters are robustly induced in a detoxification mutant by 5mM t-2-hex (Figure 3D,E) and significantly induced by 10mM t-2-hex in detoxification competent wild type cells (new Figure 3F); mitochondrial pre-protein accumulation is induced by 10mM t-2-hex in wild type cells (Figure 5G); several subunits of the TOM and TIM complexes are lipidated by low (10mM) t-2-hex doses in wild type cell extracts (Figure 8B), mitochondrial import assays with mt-GFP in intact yeast wild type cells reveal that t-2-hex significantly inhibits import at low (5mM) t-2-hex concentrations (new Suppl. Figure 1). 5-10mM t-2-hex applied here is considerably lower than the published data in human cells with ³ 25mM on intact cells or cell extracts (Jarugumilli et al. 2018).

The main claim of the manuscript that t-2-hex targets the TOM complex and inhibits mitochondrial protein import is not supported by experimental data as import was not experimentally investigated. The observed accumulation of precursor proteins could have many other reasons (e.g. dissipation of membrane potential, defects in mitochondrial presequence proteases, defects in cytosolic chaperones, modification of mitochondrial precursors by t-2-hex rendering them aggregation prone and thus non-import competent). However, none of these alternative explanations have been experimentally addressed or discussed in the manuscript.

We have now performed additional experiments, alternative to the pre-protein quantifications, showing that t-2-hex specifically inhibits mitochondrial protein import. We investigated the effect of t-2-hex on mitochondrial protein import using flow cytometric GFP assays in live yeast cells. Specifically, we compared the expression and maturation of GFP targeted either to the cytosol or the mitochondrial matrix and show that low doses of t-2-hex (≥5 μM) significantly inhibited mt-GFP activity compared to cytosolic GFP in wild-type cells (new Supplemental Figure 1B). In contrast, this inhibition was not observed with the saturated derivative, t-2-hex-H2. Flow cytometric rhodamine123 assays revealed that t-2-hex did not alter ΔΨm within the concentration range that efficiently inhibits mt-GFP activity (new Supplemental Figure 1C). Alternative t-2-hex effects such as the direct modification of mitochondrial pre-proteins or cytosolic chaperones, potentially making the precursors prone to aggregation, are less likely, as the mitochondrial and cytosolic GFP used in these import studies differ only by the small, cysteine-free PreSu9 pre-peptide. This information is now included in the Results and Discussion sections.

Furthermore, many of the results have been reported before (interaction of Tom22 and Tom70 with Hfd1) or observed before (TOM40 as target of t-2-hex in human cells).

The interaction of Tom22 or Tom70 with Hfd1 has been only reported in high throughput pull-down studies in yeast (Opalinski et al., 2018 and Burri et al., 2006), and no functional connection between Hfd1 lipid detoxification and TOM has been investigated. Here we corroborate these high throughput results by targeted pull-down experiments, which strengthens the new finding that Hfd1 functionally interacts with the TOM complex. Tom40 has been found to be lipidated by high t-2-hex concentrations in human cell extracts in high throughput in vitro proteomic studies (Jarugumilli et al., 2018), but no functional connection between human TOM and t-2-hex has been investigated so far. Here we corroborate these high throughput results by targeted experiments, which strengthens the new findings that t-2-hex and TOM interact functionally.

**Recommendations for the authors:**

**Reviewer #2 (Recommendations For The Authors):**
Congratulations on this exciting study. Even if some of the mechanistic details will have to be addressed in further studies (which of the modified sites are physiologically relevant; which sites are modified in vivo without external addition of t-2-hex) this study is inspiring and opens a new direction of mitochondrial research. I therefore fully support publication of this nice study in its current form.
**Reviewer #3 (Recommendations For The Authors):**
Two of the reviewers pointed out that the observation of precursors in whole cell extract is not sufficient to draw conclusions on mitochondrial protein import rates. The authors did not provide any new experiments but argued that a recent publication (Weidberg and Amon, 2018) had used the same readout for this conclusion. Why this manuscript was accepted with this statement is not known to this reviewer, but it does not change the fact, that the conclusion is not valid. Many alternative explanations are possible (see public review) and the claim that the import competence of the TOM complex is affected upon t-2-hex treatment is not appropriate.

We have now performed new experiments addressing the inhibition of mitochondrial protein import by t-2-hex as an alternative to our precursor accumulation assays. We compared the induced expression of cytosolic and mitochondrial GFP by flow cytometry as a quantitative mitochondrial import assay (Sirk et al., Cytometry A. 2003 Nov; 56(1) 15-22). Low doses of t-2-hex (≥5 μM) significantly inhibited mt-GFP activity as compared to cytosolic GFP in wild-type cells (new Supplemental Figure 1B). This inhibition of mitochondrial GFP is independent of mitochondrial membrane potential perturbation (new Supplemental Figure 1C) and alternative t-2-hex effects, such as the direct modification of the mtGFP precursor or cytosolic chaperones are less likely, as the mitochondrial and cytosolic GFP used in these import studies differ only by the small, cysteine-free PreSu9 pre-peptide.

The first sentence of the abstract states that t-2-hex „induces mitochondrial dysfunction in a conserved manner from yeast to human". I find two issues with this statement: (1) if the mechanism is known what is the question addressed in the present manuscript and (2) the second sentence of the results fully contradicts the above sentence „In human cells, t-2-hex causes mitochondrial dysfunction by directly stimulating Bax-oligomerisation at the outer mitochondrial membrane. In yeast, however, t-2-hex efficiently interferes with mitochondrial function and cell growth in a Bax independent manner."

We agree that the first sentence was misleading, this has been fixed now in the revised version.

The first reviewer requested a repetition of key experiments with lower concentrations and the authors provided additional in vitro data, however, for this, 10 uM is still very high. To gain valuable and physiological relevant data the initial transcriptomic analysis should be repeated with a low amount and in a wild-type yeast background.

Published t-2-hex chemoproteomic experiments on human cell extracts were performed at higher concentrations (>25mM) and human Bax is hardly lipidated by 10mM t-2-hex (Jarugumilli et al., 2018), therefore the in vitro lipidation data provided in our study should be considered a low t-2-hex dose. The initial transcriptomic study with high doses of t-2-hex in a detoxifying mutant is a starting experiment and was aimed at identifying as many determinants of t-2-hex toxicity as possible. Building on this, we further show that proteostasis and mitochondrial protein trafficking, the relevant cellular functions for our study, are physiologically important targets at low t-2-hex doses in several independent experimental approaches: proteostasis-specific gene expression is robustly induced in a detoxification mutant by 5mM t-2-hex (Figure 3D,E) and significantly induced by 10mM t-2-hex in detoxification competent wild type cells (new Figure 3F); mitochondrial pre-protein accumulation is induced by 10mM t-2-hex in wild type cells (Figure 5G); several subunits of the TOM and TIM complexes are lipidated by low (10mM) t-2-hex doses in vitro in wild type extracts (Figure 8B), mitochondrial import assays with mt-GFP in intact yeast wild type cells reveal that t-2-hex significantly inhibits import at low (5mM) t-2-hex concentrations (new Suppl. Figure 1).

As already stated above there are many alternative explanations for the observed accumulation of precursor proteins, e.g. the decreased proteasome activity could be cause and not consequence. Also, the modification of precursors directly upon translation in the cytosol could likely impact on their further transport and result in direct aggregation in the cytosol.

As mentioned above, we have now corroborated the t-2-hex specific mitochondrial protein import defect by alternative in vivo experiments, which are not dependent on the accumulation of mitochondrial precursors. We have tested now the possibility that decreased proteasome activity could indirectly inhibit mitochondrial import. This is not the case because a rpn4 mutant with impaired proteasomal activity induces normal mtGFP levels (new Suppl. Figure 1D). We cannot exclude that the modification of precursors by t-2-hex upon translation might additionally impact on the transport of some mitochondrial pre-proteins. However, mitochondrial and cytosolic GFP used in the import studies only differ in the small cysteine-free PreSu9 pre-peptide making it very unlikely that precursor lipidation is secondarily responsible for the observed import defect.

Many of the comments after first reviewing the manuscript were not addressed experimentally although many of the suggested experiments are easy to perform. I can only encourage the authors to provide more experimental support and controls, as the claims are currently not sufficiently supported.

In the two revisions of our manuscript, we have included several control experiments to better link the pro-apoptotic lipid t-2-hex with mitochondrial import stress. These include: in vitro lipidation of TOM/TIM subunits by low t-2-hex concentrations, t-2-hex tolerance and recovery of mitochondrial protein import in specific tom mutants, inhibition of mitochondrial protein import (pre-protein and mtGFP assays) by low t-2-hex doses independently on mitochondrial membrane potential and proteasome activity, and induction of proteostasis specific gene expression by low t-2-hex doses.